# MVR-cache: Optimizing Semantic Caching via Multi-Vector Retrieval and Learned Prompt Segmentation

Ali Noshad [1]    Zishan Zheng [2]    Yinjun Wu [1]

## Abstract

To reduce LLM costs and latency, semantic caching systems must accurately identify when a new prompt matches a cached one. Current methods often rely on simplistic similarity measures, which limit their effectiveness. We introduce MVR-cache, a novel semantic caching approach that significantly improves retrieval accuracy by integrating Multi-Vector Retrieval (MVR). MVR-cache is built upon a learnable segmentation model that intelligently splits prompts, enabling fine-grained similarity comparisons via MaxSim. We derive the model's training objective from a rigorous theoretical analysis. This can ensure that optimizing this objective directly maximizes cache hits under strict correctness constraints. To solve the resulting non-differentiable combinatorial optimization problem, we leverage a reinforcement learning-based training strategy with the theoretically grounded objectives as the reward. Experimental results on established benchmarks across diverse tasks confirm that in comparison to the state-of-the-art, MVR-cache consistently increases the cache hit rates by up to 37% while maintaining the same correctness guarantees. MVR-cache is available at `https://github.com/PKU-SDS-lab/MVR-Cache`

## 1. Introduction

Semantic caching has emerged as an effective mechanism for reducing the latency and computational cost of large language model (LLM) inference (Xiong et al., 2024; Schroeder et al., 2025). Traditional caches based on exact string matches fail to exploit the underlying semantic similarity between different, but conceptually equivalent, prompts—requiring redundant model invocations. To address this limitation, semantic caches can serve paraphrased or semantically related queries by embedding prompts into a semantic vector space and reusing responses based on similarity, substantially increasing the *cache hit rate, i.e., the ratio of reusing the cache's response*, and thus improving system efficiency (Bang, 2023; Dasgupta et al., 2024). Production examples include Azure semantic caching option for LLM APIs [1], and LiteLLM caching system [2]

Existing semantic caching methods primarily focus on how to design reasonable cache policies to determine when to reuse the cache or not, based on the similarity scores between incoming prompts and their nearest neighbors in the cache. For instance, the static policy approaches (Dasgupta et al., 2024; Li et al., 2024; Bang, 2023)[1][2] compare such similarity scores against a global threshold to decide the cache's reuse, while the recently proposed vCache method (Schroeder et al., 2025) employs dynamically learned, prompt-specific thresholds to provably guarantee correct response. Only when such similarity scores are high enough, the cached response of the nearest neighbors is reused. However, the similarity measures used in these methods are all simple ones, such as cosine similarity between the prompts' embeddings, which may not accurately capture the subtle semantic differences in prompts, particularly for those semantically complex ones. Consequently, semantically dissimilar prompts may be incorrectly matched, degrading the cache hit rate.

For instance, we draw a prompt, $x$, from the SemCacheClassification dataset (Schroeder et al., 2025) in Figure 1, which is a positive review of an adult crime drama. In a standard cosine similarity lookup, this prompt returns $x_1$ as the nearest neighbor. While $x_1$ is also a review for the same type of drama—sharing salient keywords like "crime" that define their primary topic—it contains negative comments. This subtle semantic difference leads to a divergent LLM response for $x_1$ compared to $x$, resulting in a cache miss despite their high topical similarity.

---

[1]School of Computer Science, Peking University, Beijing, China [2]School of Information, Renmin University of China, Beijing, China. Correspondence to: Yinjun Wu <wuyinjun@pku.edu.cn>.

*Proceedings of the 43$^{rd}$ International Conference on Machine Learning*, Seoul, South Korea. PMLR 306, 2026. Copyright 2026 by the author(s).

[1]`https://learn.microsoft.com/en-us/azure/api-management/azure-openai-enable-semantic-caching`
[2]`https://docs.litellm.ai/docs/proxy/caching`

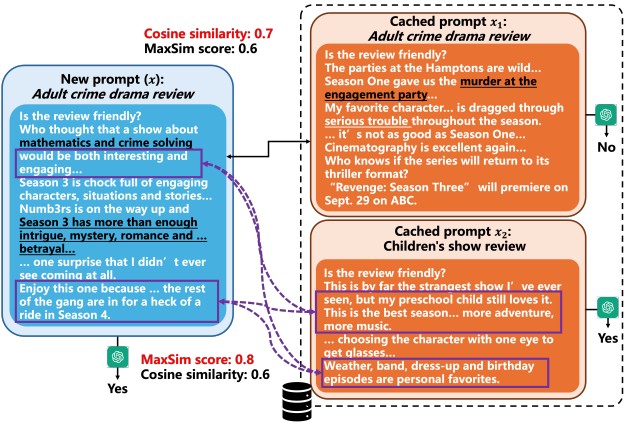

*Figure 1.* A motivating example from the SemCacheClassification dataset (Schroeder et al., 2025). Using a single embedding per prompt, a new query $x$ about whether an adult crime drama review is friendly finds its nearest neighbor ($x_1$) via cosine similarity. However, their LLM responses differ, thus causing a cache miss. In contrast, MVR-cache learns to segment prompts, particularly extracting the sentences with positive comments as individual segments (highlighted with purple boxes) for $x$ and its real nearest neighbor, $x_2$. By embedding these segments, the segmentation-aware MaxSim score of $x_2$ to $x$ exceeds that of other cached prompts, thus allowing the reuse of the cached response of $x_2$ for a cache hit.

To address this issue, we propose to employ Multi-Vector Retrieval (MVR) to facilitate the retrieval of the real nearest neighbor for a new prompt. Originally proposed for information retrieval by ColBERT (Khattab & Zaharia, 2020), MVR decomposes both queries and documents into fine-grained pieces and encodes each piece into one embedding, thus producing multi-vector representations for queries and documents. The query-document similarity is then computed using the *MaxSim score*: for each query vector, its maximum similarity with any document vector is identified, and these scores are aggregated. As revealed by ColBert (Khattab & Zaharia, 2020) and its follow-up works, MVR provides a nuanced matching mechanism that excels at capturing partial matches, thereby enhancing retrieval accuracy. However, its performance hinges critically on the strategy used to segment text since the multi-vector representations are produced by embedding each of these segments. While ColBERT's default approach uses single-token embeddings, recent work (Liu et al., 2025) has shown this to be suboptimal, demonstrating that optimized segmentation is key to unlocking MVR's full potential.

We therefore propose a novel prompt segmentation method, MVR-cache, for semantic cache, which aims to *maximize the cache hit rate while strictly adhering to user-specified error guarantees*. MVR-cache begins with the development of **a lightweight segmentation model** designed with two essential properties: (1) *minimal model capacity to ensure efficiency* so that the decision of whether to exploit the cached prompts is made instantly online, and (2) the ability to produce *a variable number of segments per prompt*, which is fundamental to the flexible nature of MVR where the vector count per sample varies. However, two critical challenges arise for effective training over this model.

First, the *cache hit rate is a discrete and non-differentiable metric*, which thus cannot serve as the training objective for the segmentation model. Instead, we propose a surrogate objective, which aims to align the segmentation-aware MaxSim score of a new prompt to its nearest neighbor in the cache with identical LLM responses. As a consequence, for the example in Figure 1, the MaxSim score between $x$ and its real nearest neighbor $x_2$ could become higher than others. A **rigorous theoretical analysis** reveals that optimizing this objective guarantees maximizing the vCache cache hit rate without violating the correctness constraints.

In addition, due to the discrete nature of segmentation decisions, the output space of the segmentation model is large and complex, which is a *combinatorial space of all possible segmentation points on each prompt*. Hence, optimizing the above objective remains *non-differentiable*. To overcome this, we frame the problem of training the segmentation model as a **reinforcement learning for combinatorial optimization (RL4CO)** (Berto et al., 2023) task, where we design a reward function directly informed by our theoretical objective, enabling effective gradient-based learning of the segmentation model.

We further perform extensive experiments on a variety of benchmark datasets, covering diverse tasks such as classification, search and open-ended generation. The results demonstrate that in comparison to the state-of-the-art, MVR-cache can consistently increase the cache hit rate by up to 25% while respecting the correctness guarantees. Our contributions are summarized as follows:

- MVR-cache, a method that learns an **efficient, lightweight segmentation model** to segment prompts to facilitate multi-vector retrieval (MVR) to identify the real nearest neighbors in the cache for a new prompt.
- A **rigorous theoretical analysis** that formalizes a training objective, proving that optimizing it is equivalent to maximizing cache hit rate under correctness guarantees.
- A **reinforcement learning (RL) solution** to train this segmentation model by framing it as a **combinatorial optimization problem**.
- **Extensive experiments** on a variety of benchmark datasets covering diverse tasks demonstrate the effectiveness of MVR-cache.

## 2. Preliminaries

### 2.1. Semantic Caching

Let $x_1, \ldots, x_n$ be prompts sequentially inserted into a semantic cache. For each $x_i$, mainstream solutions like vCache (Schroeder et al., 2025) store three items: its embedding $\mathcal{E}(x_i) \in \mathbb{R}^d$, its LLM-generated response $r(x_i) = \text{LLM}(x_i)$, and associated metadata $\mathcal{O}(x_i)$. $\mathcal{O}(x_i)$ is a sequence of pairs, where each pair links $x_i$ to a later prompt $x_j$ that identified $x_i$ as its nearest neighbor. Formally:

$$\mathcal{O}(x_i) = \{(s(x_j), c(x_j)) | nn(x_j) = x_i\}_{j=i+1}^n, \quad (1)$$

in which $nn(x_j)$ denotes the nearest neighbor of $x_j$ among $\{x_1, x_2, \ldots, x_{j-1}\}$, $s(x_i)$ is defined as the similarity score between $x_j$ and $nn(x_j)$, and a Boolean label $c(x_j)$ indicating whether the true response $r(x_j)$ matches the cached response $r(nn(x_j))$.

The goal of performing semantic caching is to reduce LLM inference costs by reusing cached responses for new, sufficiently similar prompts. The core of this approach is to *develop a caching policy* that decides, for each new query, whether to *exploit* (reuse a cached response) or *explore* (invoke the LLM).

### 2.2. Caching policy in vCache

To achieve the above goal, vCache (Schroeder et al., 2025) proposes a *stochastic caching policy* based on the similarity of one new prompt, $x$, to its nearest neighbor in the cache, $s(x)$. This policy aims to maximize the cache hit rate while ensuring that the probability of returning the correct response is provably high enough. Due to the dependency of this policy on $s(x)$, vCache begins by modeling the conditional *correctness probability* that the LLM response of a new prompt, $x$, matches that of its nearest neighbor given $s(x)$ using the following sigmoid formula:

$$\Pr\big(c(x) = 1 \mid s(x)\big) = \mathcal{L}(s(x); t, \gamma) = \frac{1}{1 + e^{-\gamma(s-t)}}. \quad (2)$$

In the above formula, $t \in [0, 1]$ and $\gamma > 0$. These two parameters are *dependent on specific cached prompts*, and estimated via maximum likelihood estimation (MLE) using all the metadata of the nearest neighbor of $x$, $\mathcal{O}(nn(x))$:

$$(\hat{t}, \hat{\gamma}) = \arg\min_{t, \gamma} \sum_{(s_i, c_i) \in \mathcal{O}(nn(x))} \ell_{\text{BCE}}(\mathcal{L}(s_i; t, \gamma), c_i), \quad (3)$$

in which, $\ell_{\text{BCE}}$ represents the binary cross-entropy loss. These two estimated parameters are then incorporated into Equation (2) to estimate the correctness probability for each new prompt $x$. This probability is then utilized to dynamically determine the exploration probability, $\tau$, which governs when to bypass the cache and query the LLM directly.

Specifically, to compensate for potential overfitting of the parameters $(\hat{t}, \hat{\gamma})$ in Equation (3), which could lead to *an overestimate of* $\Pr\big(c(x) = 1 \mid s(x)\big)$, $\tau$ is conservatively estimated as:

$$\tau = \min_{t', \gamma'} \frac{(1 - \delta) - \alpha}{1 - \alpha}, \quad \text{with } \alpha = (1 - \epsilon) \mathcal{L}(s; t', \gamma'), \quad (4)$$

where the minimization is over $(t', \gamma')$ within a $(1 - \epsilon)$ confidence region of $(\hat{t}, \hat{\gamma})$. This formulation guarantees, under mild assumptions, **the probability of a correct response (via cache or LLM) is at least $1 - \delta$ for any user-defined error rate $\delta$** (see (Schroeder et al., 2025) for derivation).

### 2.3. Multi-Vector Retrieval

As noted in Section 1, cosine similarity between full-prompt embeddings may not reflect true semantic similarity, leading to suboptimal cache hit rates.

To address this issue, we employ multi-vector retrieval (MVR) (Khattab & Zaharia, 2020), which represents each text sequence with a set of embedding vectors rather than a single vector. Specifically, let a cached prompt $x_j$ and a new prompt $x$ be represented by sequences of embeddings $\{\mathcal{E}(x^{(1)}), \mathcal{E}(x^{(2)}), \ldots\}$ and $\{\mathcal{E}(x_j^{(1)}), \mathcal{E}(x_j^{(2)}), \ldots\}$. Their similarity is then measured by the following *MaxSim* score:

$$\text{MaxSim}(x, x_j) = \sum_t \max_s \text{sim}(\mathcal{E}(x^{(t)}), \mathcal{E}(x_j^{(s)})). \quad (5)$$

Intuitively speaking, $\text{MaxSim}(x, x_j)$ first selects the most similar vector, $\mathcal{E}(x_j^{(s)})$, from $x_j$ for each $\mathcal{E}(x^{(t)})$, and then aggregates this similarity across all $t$.

*Example* 2.1. Suppose one cached prompt $x_1$ is represented by three vectors, $[x_1^{(1)}, x_1^{(2)}, x_1^{(3)}]$ while the new prompt $x$ is associated with two vectors, $[x^{(1)}, x^{(2)}]$, we compute the pairwise similarity scores between each vector from $x_1$ and each vector from $x$ as follows:

|           | $x_1^{(1)}$ | $x_2^{(1)}$ | $x_3^{(1)}$ |
|-----------|-------------|-------------|-------------|
| $x^{(1)}$ | 0.01        | 0.83        | 0.02        |
| $x^{(2)}$ | 0.05        | 0.80        | 0.01        |

For each $x^{(t)}$ from $x$, we identify its most similar vector among $[x_1^{(1)}, x_1^{(2)}, x_1^{(3)}]$ and highlight its similarity to $x^{(t)}$ using red color, which is the same as the highest similarity score at each row. These scores are then plugged into Equation (5), yielding the final MaxSim score $\text{MaxSim}(x, x_1) = 0.83 + 0.80 = 1.63$

In Multi-Vector Retrieval (MVR), the effectiveness of MaxSim scores depends critically on *the strategy used to decompose text into segments*. While existing methods like ColBERT (Khattab & Zaharia, 2020) generate embeddings at the token level, this overly fine-grained approach can impair retrieval performance. To address this, (Liu et al., 2025) introduces a method that dynamically decomposes text into coarser, semantically meaningful segments. This approach uniquely leverages retrieval performance as an optimization signal to iteratively refine the decomposition process.

**Problem definition**  Building on this insight, we adapt the strategy of (Liu et al., 2025) to semantic caching. Our method *automatically decomposes both cached and new prompts*, then uses a *segmentation-aware MaxSim score* to more reliably retrieve the real nearest neighbors. The

ultimate goal is to *maximize the cache hit rate while maintaining the correctness guarantees*.

## 3. MVR-cache

### 3.1. General inference process

The overall inference process of MVR-cache is visualized in Figure 2, which is described in detail below.

**Prompt segmentation and embedding**  Given an arbitrary text sequence $x$—including both incoming new prompts and cached prompts—MVR-cache first identifies a set of candidate split positions, e.g., punctuation boundaries. These split positions are denoted by $\mathcal{P}_x = \{p_1, p_2, \ldots, p_i, \ldots\}$. A segmentation model $\Theta$ then selects a subset of these positions to split the sequence. Formally, the model outputs an ordered list of split indices:

$$\Theta(x) = \overrightarrow{p} = [p_{i_1}, \ldots, p_{i_{m-1}}], 1 \leq i_1 < \cdots < i_{m-1} < |\mathcal{P}_x|,$$

Using these indices, $x$ is partitioned into $m$ contiguous subsequences, $(x^{(1)}, \ldots, x^{(m)})$, such that each $x^{(t)}$ consists of tokens from position $(p_{i_t}) + 1$ to position $(p_{i_{t+1}})$ in $x$, with the conventions $p_{i_0} = 0$ and $p_{i_m} = |x|$.

Each $x^{(t)}$ is then embedded using a shared encoder $\mathcal{E}(\cdot)$, producing $m$ vectors, $(\mathcal{E}(x^{(1)}), \mathcal{E}(x^{(2)}), \ldots, \mathcal{E}(x^{(m)}))$ as the multi-vector representation for $x$. The overall segmentation and embedding process can be summarized as follows:

$$\mathbf{SEG}(x; \Theta(x)) = (x^{(1)}, \ldots, x^{(m)})$$
$$\mathbf{Emb}((x^{(1)}, \ldots, x^{(m)})) = (\mathcal{E}(x^{(1)}), \ldots, \mathcal{E}(x^{(m)})) \quad (6)$$

*Example* 3.1. For instance, $\mathcal{P}_x$ could be defined as an ordered sequence of positions where punctuation marks appear in $x$. For the following prompt $x$, $\mathcal{P}_x$ =[6, 14, 28] since the comma occurs at the token position 6 and 14, and a period occurs at the token position 28. The period at the final position may also be treated as a special "<stop>" token.

```
x = ``Summarize Section 3, list three
    limitations, and format as bullet points.''
```

Given $\mathcal{P}(x)$, a segmentation model $\Theta$ may output a subset of these positions to indicate where the prompt should be divided. For instance, $\Theta$ might output [14], indicating that the second comma at position 14 is used to split $x$ into two sub-subsequences:

```
x^(1) = ``Summarize Section 3, list three
        limitations,''
x^(2) = ``and format as bullet points.''
```

Note that $\mathcal{P}_x$ is prompt-dependent, which has *variable sizes* between different prompts. Similarly, the output of $\Theta$ is a variable-length subset of $\mathcal{P}_x$.

**Segmentation-aware MaxSim score**  The above segmentation-and-embedding procedure can generate a

multi-vector representation for each cached prompt $x_i$ and each new prompt $x$:

$$(\mathcal{E}(x_i^{(1)}), \mathcal{E}(x_i^{(2)}), \cdots) = \mathbf{Emb}(\mathbf{SEG}(x_i; \Theta(x_i)))$$
$$(\mathcal{E}(x^{(1)}), \mathcal{E}(x^{(2)}), \cdots) = \mathbf{Emb}(\mathbf{SEG}(x; \Theta(x)))$$

These multi-vector representations can then be incorporated into Equation (5) to compute the MaxSim score between each $x_i$ and $x$. However, the MaxSim score is inherently *asymmetric*, which poses a critical issue in semantic caching. Returning to Example 2.1, MaxSim$(x_1, x)$ is computed by aggregating the highest score at each column, resulting in a score of 0.89, which is only roughly half of MaxSim$(x, x_1)$. In semantic caching, a cache hit should reflect *mutual semantic similarity* between the new and cached prompts. A high MaxSim score, which is unidirectional, however, can only guarantee partial matching between prompts-specifically, that each segment of the new prompt is similar to some segment of the cached prompt, but not necessarily vice versa. Hence, we propose to average the two normalized unidirectional MaxSim scores as the **Segmentation-aware MaxSim score** for semantic caching:

$$\text{SMaxSim}_\Theta(x_i, x_j)$$
$$:= 0.5 \times \left[ \frac{1}{|x_i|} \text{MaxSim}(x_i, x_j) + \frac{1}{|x_j|} \text{MaxSim}(x_j, x_i) \right] \quad (7)$$

where $|x_i|$ and $|x_j|$ denote the number of segments in each prompt, and normalization ensures scale invariance across prompts of different lengths. It is parameterized by $\Theta$ since it relies on the segmentation produced by the model $\Theta$. This symmetric measure better captures bidirectional semantic relevance, making it more suitable for cache hit determination. Using SMaxSim$_\Theta$, the nearest neighbor for a new prompt $x$ is identified as

$$s_\Theta(x) = \text{SMaxSim}_\Theta(x, nn_\Theta(x))$$
$$nn_\Theta(x) = \text{argmax}_{x_j} \text{SMaxSim}_\Theta(x, x_j).$$

In the above formula, $s_\Theta(x)$ and $nn_\Theta(x)$ are parameterized by $\Theta$ due to its reliance on the segmentation model, $\Theta$. During the inference phase, $nn_\Theta(x)$ can be quickly retrieved using indexes constructed for multi-vector retrieval, e.g., PLAID (Santhanam et al., 2022). The corresponding similarity score, $s_\Theta(x)$, is then passed to the vCache module described in Section 2.2 (Equations (2)–(4)) to determine the exploration probability $\tau$ for prompt $x$. Note that according to Equation (3), $\hat{t}$ and $\hat{\gamma}$ also implicitly depend on $\Theta$, which are thus dynamically updated during the training process as $\Theta$ evolves (see Section 3.4).

### 3.2. Segmentation model design

**Lightweight and Variable-Size Segmentation Model.** The segmentation model $\Theta$ is a critical component for ensuring segmentation quality and the quality of subsequent multi-vector representations. Ideally, $\Theta$ should possess two key properties. First, it must be sufficiently *lightweight*—ideally, much smaller than the underlying LLM—to enable real-time segmentation of new prompts during online inference. If $\Theta$ were comparable in size to the LLM, processing each

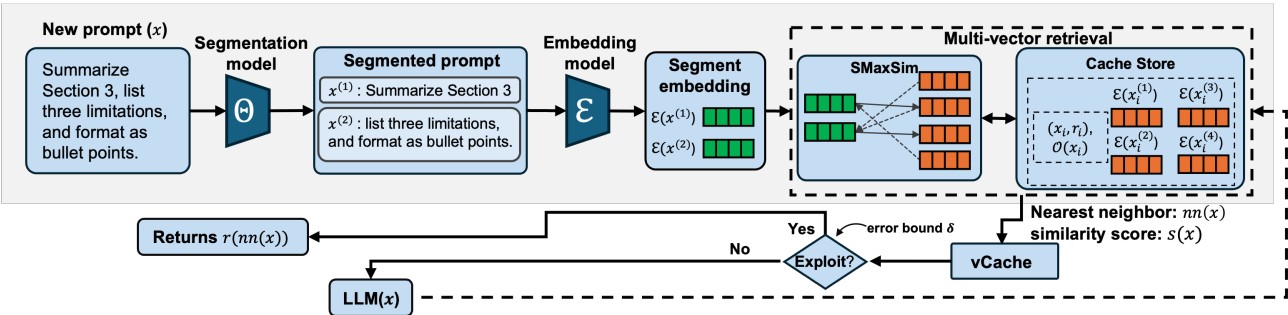

*Figure 2.* Overview of the inference process for MVR-cache. Given a new prompt $x$, the segmentation model $\Theta$ first identifies candidate split positions (e.g., at punctuation marks) to yield multiple text segments. Each segment is encoded by the embedding model $\mathcal{E}$ to produce a multi-vector representation for $x$. This representation is compared to cached entries using the segmentation-aware MaxSim score (SMaxSim) to retrieve the nearest neighbor $nn(x)$ and its similarity score. Finally, the vCache module uses $nn(x)$ and its SMaxSim score to determine whether to utilize the cached prompt.

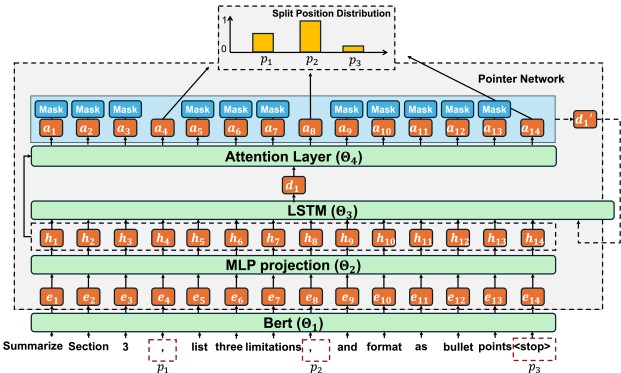

*Figure 3.* Model architecture for prompt segmentation. This segmentation model encodes input prompt tokens using a BERT encoder ($\Theta_1$). A projection MLP ($\Theta_2$) transforms the token embeddings, which are then processed by a decoder LSTM ($\Theta_3$). Finally, an attention layer ($\Theta_4$) computes the probability distribution over candidate split positions to predict segment boundaries.

prompt would incur near-LLM-level computational overhead, negating the acceleration benefits of caching. Second, $\Theta$ must accommodate *variable-length inputs and outputs*, as both the prompt and the number of segments are dynamic.

To fulfill these requirements, we design a lightweight model based on the pointer network architecture (Vinyals et al., 2015), which naturally handles sequences of variable sizes (see Figure 3). Given an input prompt $x$ of length $L$, our model first encodes $x$ using a BERT encoder $\Theta_1$ to obtain token embeddings $(e_1, \ldots, e_L)$. A single-layer MLP encoder $\Theta_2$ then transforms each $e_i$ into a pointer state $h_i$ for $i = 1, \ldots, L$. These states are aggregated into an initial hidden state $d_1$ via a single-layer LSTM $\Theta_3$:

$$d_1 = \text{LSTM}_{\Theta_3}([h_1, h_2, \ldots, h_L]).$$

The resulting $d_1$ represents the entire prompt context. Following the pointer network mechanism, an attention layer

$\Theta_4$ uses $d_1$ and the pointer states $h_i$ to compute a probability distribution over all potential split positions. However, unlike the standard pointer network which considers all input indices, our candidate split positions are a restricted subset $\mathcal{P}_x$ of $[1, \ldots, L]$. We therefore mask invalid positions by assigning them zero probability. The attention mechanism for this step is formulated as:

$$u_{1j} = v^\top \tanh(W_1 h_j + W_2 d_1), j \in [1, 2, \ldots, L]$$
$$a_{1j} = \text{softmax}(u_{1j}) \cdot \mathbf{I}(j \in \mathcal{P}_x), d_1' = \sum_j a_{1j} h_j \quad (8)$$

where $v$, $W_1$, and $W_2$ are learnable parameters in the attention layer. Hence, $\Theta_4 = (v, W_1, W_2)$. The position $j$ with the highest probability $a_{1j}$ is selected as the first segmentation boundary.

Overall, the segmentation model consists of one Bert encoder ($\Theta_1$), one MLP layer ($\Theta_2$), one single-layer LSTM ($\Theta_3$) and one simple attention layer ($\Theta_4$). In our implementation, $\Theta$ only takes around 500-600 MB GPU memory in total, which is much smaller than the capacity of typical LLMs.

**Recurrent selections of the split positions**    To select subsequent boundaries, this process proceeds recurrently. The attention output $d_i'$ is fed back into the LSTM, $\Theta_3$, to update the context state for the next step:

$$d_2 = \text{LSTM}_{\Theta_3}([h_1, h_2, \ldots, h_L, d_1']). \quad (9)$$

We then compute a new distribution by replacing $d_1$ with $d_2$ in Equation 8. To prevent reselection, all candidate positions up to and including the previously chosen index are masked. This recurrent selection continues until the termination token ($<$stop$>$) is predicted.

## 3.3. Theoretical insights for the training objectives

Recall that our goal is to *maximize cache hit rate* while preserving vCache's *correctness guarantee* through appropriate prompt segmentation. To this end, we adapt the segmentation-aware MaxSim score, $\text{SMaxSim}_\Theta$, by training the segmentation model to minimize the MLE loss in Equation (3). This aligns the similarity scores $\text{SMaxSim}_\Theta(x_i, x_j)$ with the binary label indicating whether $x_i$ and $x_j$ yield equivalent LLM responses.

However, as mentioned in Section 3.1, a complication arises because the optimal $t$ and $\gamma$ implicitly depend on $\Theta$, thus leading to a complex dependency of the final cache hit rate on $\Theta$. Nevertheless, we can formally show that under the following assumptions, optimizing the MLE loss maximizes the hit rate while maintaining the correctness guarantee.

**Assumption 3.1** (Conditional distribution of similarity scores)**.** The similarity score $s(x)$, conditioned on the annotation label $c$, follows a normal distribution: $\Pr(s \mid c) \sim \mathcal{N}(\mu_c, \sigma^2)$, where $\mu_c$ differs between different classes, $c$ (could be either 0 or 1).

**Assumption 3.2** (Balanced class prior)**.** The class distribution is balanced: $\Pr(c = 1) = \Pr(c = 0) = 0.5$.

**Theorem 3.3.** *Under Assumptions 3.1 and 3.2, optimizing the prompt segmentation model to minimize Equation (3) maximizes the cache hit rate under an arbitrary user-specified error bound $\delta$.*

In practice, the balanced-class assumption (3.2) often does not hold. We therefore employ a class rebalanced version of Equation (3). The following lemma extends our guarantee to this setting.

**Lemma 3.4.** *Under Assumption 3.1 alone, minimizing the class-rebalanced version of Equation (3) increases the cache hit rate under the same error bound $\delta$.*

We defer the full proofs of Theorem 3.3 and Lemma 3.4 to Appendix A. We include some empirical results in Appendix D that can justify Assumption 3.1 on real datasets.

## 3.4. Offline Training of the Segmentation Policy

Following the above theoretical analysis, the training objective is to minimize the following MLE loss over all training prompts $x_i$ and their associated nearest neighbors $x_j$:

$$\sum_i \sum_{nn_\Theta(x_j)=x_i} \ell_{\text{BCE}}(\mathcal{L}(\text{SMaxSim}_\Theta(x_i, x_j); t_i, \gamma_i), c_j)$$

where $nn_\Theta(x_i)$ denotes the cached prompt most similar to $x_i$ under the segment-aware similarity measure $\text{SMaxSim}_\Theta$. Optimizing this objective requires adapting the segmentation model $\Theta$ to select an optimal set of split indices from the candidate set $\mathcal{P}_x$ for any prompt $x$. However, this poses two critical training challenges: 1) **Combinatorial Optimization**: the search space consists of all possible combinations of split indices in $\mathcal{P}_x$, which grows exponentially with the prompt length; 2) **Non-Differentiability**: the output of $\Theta$—a discrete set of indices—does not carry gradients, preventing the direct use of gradient-based optimization.

To overcome these challenges, we frame the problem as a **Reinforcement Learning for Combinatorial Optimization (RL4CO)** task (Berto et al., 2023) and model $\Theta$ as a stochastic policy $\pi_\Theta$.

**Problem Formulation as RL4CO** In the RL4CO formulation, an input prompt $x$ represents the *state*. The *action* to take from this state is a candidate segmentation, defined by a sampled subset of split indices $\overrightarrow{p}$ from the candidate set $\mathcal{P}_x$. The segmentation model $\Theta$ serves as the *policy*, defining a probability distribution $\pi_\Theta(\overrightarrow{p} \mid x)$ over all possible segmentations (actions) conditioned on the input prompt.

**Reward Design and Optimization Objective** For each training step, we randomly sample a prompt $x_i$ and consider all prompts $x_j$ where $nn_\Theta(x_j) = x_i$. We then sample segmentations: $\overrightarrow{p_{x_i}} \sim \pi_\Theta(p \mid x_i)$ and $\overrightarrow{p_{x_j}} \sim \pi_\Theta(p \mid x_j)$ for $x_i$ and each $x_j$. Using these segmentations, we compute the segment-aware similarity $\text{SMaxSim}_\Theta(x_i, x_j)$ and the corresponding Binary Cross-Entropy (BCE) loss. The reward is the negative sum of these losses:

$$\text{Reward} = \sum_{nn_\Theta(x_j)=x_i} -\ell_{\text{BCE}}(\mathcal{L}(\text{SMaxSim}_\Theta(x_i, x_j); t_i, \gamma_i), c_j)$$

We optimize $\Theta$ with *REINFORCE* (Williams, 2004) by maximizing the following expected reward:

$$\max_\Theta \; \mathbb{E}_{\overrightarrow{p_{x_i}} \sim \pi_\Theta(\cdot|x_i), \overrightarrow{p_{x_j}} \sim \pi_\Theta(\cdot|x_j), \text{ for all } nn(x_j)=x_i} \left[\text{Reward}\right].$$

$$(10)$$

The expectation is approximated via Monte Carlo sampling (Sutton & Barto, 2005) from the policy $\pi_\Theta$.

**Jointly Optimizing $t_i$ and $\tau_i$** As discussed in Section 3.1, the values of $t_i$ and $\tau_i$ used in the reward depend implicitly on the current segmentation policy $\Theta$. Therefore, at each training step, we temporarily freeze $\Theta$ and update $t_i$ and $\tau_i$ by solving the maximum likelihood estimation in Equation (3) over $x_i$ and its current set of nearest neighbors.

**Training Efficiency Considerations** While the segmentations for $x_i$ and its neighbors $x_j$ are generated online at each training step, the nearest neighbor mapping $nn_\Theta(\cdot)$ itself is also dependent on $\Theta$. Ideally, updating $\Theta$ would require recalculating this mapping for the entire cache at every step, as segmentations for all cached prompts would change. However, performing this full-neighbor recalculation online in every iteration is prohibitively expensive. To resolve this efficiency bottleneck, we *keep the nearest neighbor mapping fixed* at each training step and only update it

**Algorithm 1** Training $\Theta$ via offline RL

---

**Input:** A training set of prompts with ground-truth LLM responses $\mathcal{T}$; The policy model $\pi_\theta(\cdot \mid x)$
**for** $t = 1$ **to** $T$ **do**
    Sample a prompt $x_i$ from $\mathcal{T}$, determine all $x_j$'s in $\mathcal{T}$ satisfying $nn(x_j) = x_i$
    Sample segmentation $\overrightarrow{p}$ from $\pi_\Theta(\cdot|x)$ for each $x = x_i$ and $x_j$'s satisfying $nn(x_j) = x_i$
    Perform inference on $x_i$ and each $x_j$
    Compute SMaxSim$_\Theta(x_i, x_j)$ between $x_i$ and each $x_j$.
    Update $\Theta$ by optimizing Equation (10) with REINFORCE
    Update $t_i$ and $\gamma_i$ by solving Equation (3) for $x_i$
    **if** $t\%K == 0$ **then**
        Segment all prompts in $\mathcal{T}$ using $\Theta$ and update the set of prompts taking $x_i$ as the nearest neighbor
    **end if**
**end for**

---

periodically—once every $K$ steps—where $K$ is a hyperparameter. This strategy decouples the costly neighbor search from the frequent policy updates, achieving an effective balance between model performance and computational cost.

**Training data.** Following (Schroeder et al., 2025), we assume access to ground-truth LLM responses for all training prompts. The label $c_j$ for a pair of prompts $(x_i, x_j)$ is determined via exact string matching of their corresponding LLM responses.

## 4. Experiments

### 4.1. Experimental setup

**Baseline** We compare MVR-cache against the state-of-the-art semantic caching method **vCache**(Schroeder et al., 2025), which provides a correctness guarantee on its caching policy. In addition, we attempt to incorporate the two representative segmentation methods used in MVR, **ColBert**(Khattab & Zaharia, 2020) and **POQD**(Liu et al., 2025), into vCache. The details of adapting these two methods to vCache are provided in Appendix B. Detailed configurations of MVR-cache are provided in Appendix B.

**Prompt-insertion protocols** We evaluate semantic caching under two prompt-insertion protocols. Our default setting follows the standard vCache protocol (Schroeder et al., 2025), which we denote as *cache-on-miss only*: an incoming prompt is inserted into the cache only when it results in a cache miss, while prompts served by cache hits are not inserted. This protocol is used for all main cache hit rate, error rate, and latency results. We also evaluate a *always-cache* protocol, where every incoming prompt is inserted into the cache regardless of whether it is a hit or miss. This setting keeps the cache contents identical across methods and isolates the effect of retrieval quality from differences in cache growth.

**Datasets** We evaluate MVR-cache and baseline methods on the following four datasets, covering a variety of tasks.

- **SemCacheSearchQueries**: a subset of 150K prompts from the ORCAS dataset (Craswell et al., 2020) for the *web search* task.
- **SemCacheClassification**: a benchmark containing 45K short prompts for *classification*, collected from three diverse ecommerce text classification dataset (Talmor et al., 2019; Ni et al., 2019).[1]
- **PromptBench**: PromptBench (Zhu et al., 2023) provides a framework to comprehensively evaluate the robustness of LLMs by perturbing prompts. We follow (Zhu et al., 2023) to perturb each prompt of the SQUAD-V2 (Rajpurkar et al., 2018) dataset in different manners such that perturbed prompts can ideally have the same LLM responses to the original ones. We follow (Schroeder et al., 2025) to determine the equivalence of the LLM responses between prompts. This finally yields 38K prompts for the *question answering* task.
- **QNLI:** is another *question answering* benchmark (Wang et al., 2018). We follow the same procedure as above to perturb prompts and generate labels, which results in 29K prompts.

While the first two datasets have been used to evaluate vCache (Schroeder et al., 2025), they consist primarily of short prompts that do not reflect more complex, emerging scenarios—such as multi-turn conversations or reasoning tasks with lengthy, semantically rich prompts. To address this, we include the PromptBench and QNLI datasets, which capture these previously overlooked settings.

Unlike prior systems such as vCache, our approach involves offline training of a segmentation model. Accordingly, we split each dataset into training, validation, and test subsets. The model is trained on the training split, with the validation split used for model selection. Notably, *the training split is intentionally limited—containing only 3K prompts per dataset*—to reflect the practical constraints of obtaining labeled data for this task. Indeed, according to the ablation studies in Appendix D, a training set of 3K is sufficient to train the segmentation model. To ensure fair comparison with baseline methods like vCache, all approaches are evaluated on the same set of test prompts.

**Embedding models and LLMs** Throughout the experiments, the BGE model (Chen et al., 2024) is configured as the default embedding model to embed prompt segments, while GPT-4o-mini (Hurst et al., 2024) is employed to generate the ground-truth response for each prompt.

---

[1] https://www.kaggle.com/datasets/saurabhshahane/ecommerce-text-classification

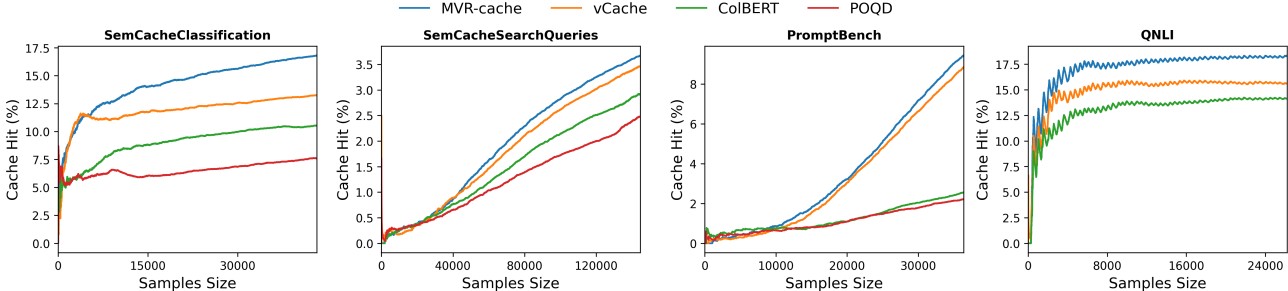

*Figure 4.* The cumulative cache hit rate VS increasing number of incoming prompts under the *cache-on-miss* protocol with $\delta = 0.01$

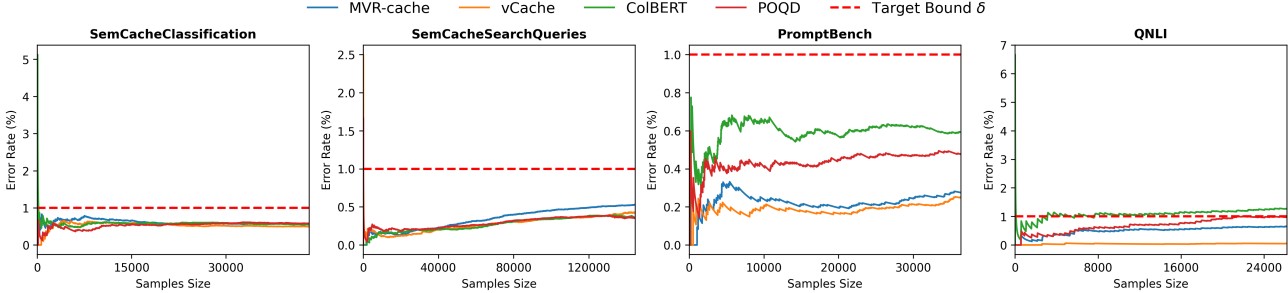

*Figure 5.* The cumulative error rate VS increasing number of incoming prompts under the *cache-on-miss* protocol with $\delta = 0.01$

**Metrics** We report the following metrics: 1) *cache hit rate*, defined as the ratio of the prompt that exploits the cache to the total number of prompts; 2) *error rate*, calculated as the ratio of false positive cache hits to the total number of new prompts; 3) *latency*, which measures the overall inference time, including the segmentation time, embedding time, retrieval time, and LLM invocation time (if cache miss).

### 4.2. Experimental results

Due to space limits, we report results for $\delta = 0.01$; the results for other $\delta$'s are in Appendix D.

**MVR-cache achieves higher cache hit rates under the same error bound** We follow (Schroeder et al., 2025) to plot the curve of the cumulative cache hit rate as the number of incoming prompts increases, which is shown in Figure 4. We also examine the real error rate in this setting. As reported in Figure 5, the error rate of all methods, including MVR-cache, gradually increases to a stable value below the user-specified $\delta$, indicating that MVR-cache respects the correctness guarantees.

The similar performance pattern indeed occurs under the *always-cache* protocol. As shown in Figures 7 and 20 (see Appendix D), MVR-cache maintains much higher cache hit rates while keeping the error rate below the user-specified bound under this protocol. Figure 7 suggests that MVR-cache can consistently achieve higher cache hit rates than all baseline methods, where the gains are up to 37% (see the

results of SemCacheSearchQueries dataset on *always-cache* protocol).

**MVR-cache reduces the end-to-end inference time.** Although MVR-cache introduces additional computation from prompt segmentation and multi-vector retrieval, it achieves higher cache hit rates, which reduces costly LLM invocations and accelerates end-to-end inference. As shown in Table 1 (under the default *cache-on-miss only* protocol), MVR-cache reduces the inference overhead by up to 6% under the same error bound, while its algorithmic overhead excluding LLM calls remains small compared with the dominant LLM inference cost. In contrast, POQD incurs substantially higher latency because it requires an additional LLM to segment each prompt, making it less suitable for time-sensitive semantic caching.

**MVR-cache generalizes across different datasets** Unlike vCache, which is a training-free method, MVR-cache requires performing additional training to obtain one segmentation model for prompt segmentation, which is conducted per dataset in the above experiments. However, we additionally evaluate the generalizability of the segmentation model trained on the PromptBench dataset to the QNLI dataset, which is reported in Figure 6. This figure surprisingly reveals that our segmentation model still outperforms all baseline methods in terms of the cache hit rate, even in the out-of-the-distribution setting.

*Table 1.* Cumulative end-to-end inference latency (minutes). Values in parentheses denote algorithm running time excluding LLM calls.

| | SemCacheClassification | SemCacheSearchQueries | PromptBench | QNLI |
|---|---|---|---|---|
| vCache | 408.49 (23.21) | 6361.52 (69.77) | 1870.57 (19.58) | 1536.00 (14.10) |
| ColBert | 501.46 (25.84) | 6521.89 (130.00) | 2294.38 (150.32) | 1626.37 (39.28) |
| POQD | 971.51 (492.92) | 6990.08 (628.33) | 2945.20 (959.60) | 2648.80 (1048.48) |
| MVR-cache | **383.32** (34.14) | **6345.61** (111.26) | **1866.58** (27.49) | **1504.43** (17.62) |

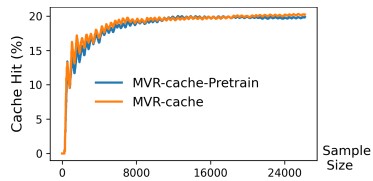

*Figure 6.* Generalization of the segmentation model trained on Prompt-Bench to QNLI.

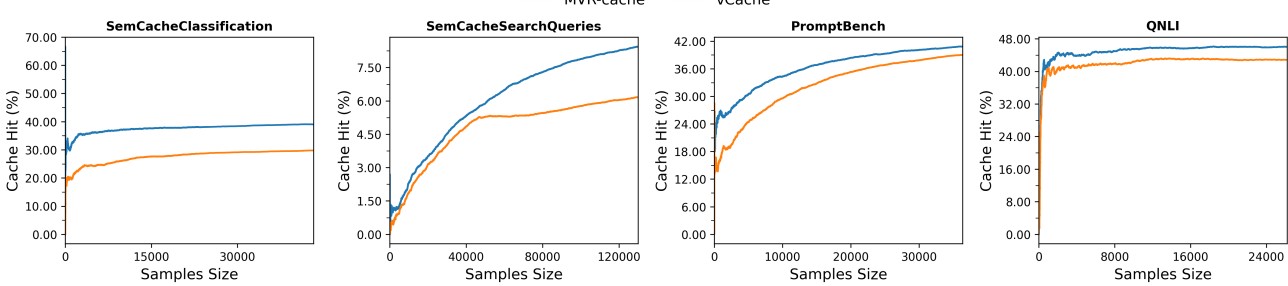

*Figure 7.* Cumulative cache hit rate e VS increasing number of incoming prompts under the *always-cache* protocol with $\delta = 0.01$.

**Training-label cost** MVR-cache uses labeled prompt pairs for offline training, collecting ground-truth LLM responses for 3K prompts per dataset this introduces a one-time labeling cost. This one-time cost is outweighed by downstream benefits: on SemCacheClassification, MVR-cache increases cache hits by 9% over vCache, saving roughly 4.1K LLM calls, already exceeding the training-label cost. Appendix D.3 shows that increasing the training set yields minimal additional gains, indicating data-efficient segmentation. We also evaluate a weak-supervision variant that uses GPT-4o-mini as a proxy labeler for GPT-4 outputs and queries GPT-4 only when proxy confidence is low. On SemCacheClassification, this avoids 80.4% of GPT-4 label calls while maintaining 97.1% agreement on proxy-labeled samples, suggesting that labeling cost can be substantially reduced without changing the training framework.

**Additional experimental results.** Further analyses, including ablations on the embedding model, training set size, and a detailed per-prompt online overhead breakdown, are provided in Appendix D.

## 5. Related work

**Semantic Caching** Current efforts to improve semantic cache hit rates largely focus on optimizing caching policies through similarity measures between prompts. While some works fine-tune embedding models to align prompts and responses (Zhu et al., 2024; ZHANG et al., 2023), these approaches lack generalizability to closed-source models and are susceptible to distribution shifts (Hajipour et al., 2022). Hence, the state-of-the-art methods avoid model tuning and instead rely on prompt similarity. These include static threshold policies (Dasgupta et al., 2024; Bang, 2023) [1] [2] and adaptive approaches like vCache (Schroeder et al., 2025), which learns prompt-specific thresholds online with correctness guarantees. *In contrast, our work is the first to enhance cache hit rates by improving the similarity measure itself, integrating multi-vector retrieval (MVR) and MaxSim.*

**Multi-Vector Retrieval** Multi-vector retrieval (MVR) overcomes the representational limitations of single-vector dense retrieval by encoding queries and documents as sets of vectors, scored via the MaxSim operator from ColBERT (Khattab & Zaharia, 2020). While subsequent work has primarily focused on improving the efficiency of MVR systems (Santhanam et al., 2021; 2022; Gao et al., 2021; Li et al., 2022), recent research (Liu et al., 2025) demonstrates the critical impact of query segmentation granularity on performance and introduces an adaptive segmentation strategy guided by retrieval accuracy. *Diverging from this approach—which uses heuristic document segmentation—our method (MVR-cache) jointly optimizes the segmentation of both new queries and cached documents.* Additional query reformulation work is discussed in Appendix C.1.

## 6. Conclusion

We present MVR-cache that integrates MVR with a learnable segmentation model. We derive a training objective that maximize hit rates while preserving correctness guarantees and formulate segmentation as a reinforcement learning for the combinatorial optimization problem.

## Acknowledgements

This work is supported by "The Fundamental Research Funds for the Central Universities, Peking University".

## Impact Statement

This paper presents work whose goal is to advance the field of machine learning. There are many potential societal consequences of our work, none of which we feel must be specifically highlighted here.

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

# A. Proof

## A.1. Preliminary for the Proof

**Definition A.1** (Two similarity metrics). Let $s_{\text{vcache}}$ be the similarity score between one prompt and its nearest neighbor in the cache store using the conventional SVR-based cosine similarity score, $s_{\text{vcache}}(\cdot)$. Likewise, let $s_{\text{MVR-cache}}$ be the similarity score under the learned similarity metric $s_{\text{new}}(\cdot)$ in MVR-cache. Both are assumed to be random variables in $[0, 1]$. We write $s_m$ for $m \in \{\text{vcache}, \text{MVR-cache}\}$.

**Definition A.2** (Class-conditional probability distribution). Assume each $s_m$ admits class-conditional densities on $[0, 1]$:

$$f_1(s) := f(s \mid C = 1), \qquad f_0(s) := f_m(s \mid C = 0),$$

and corresponding CDFs $f_1, f_0$. The marginal density is the mixture

$$p(s) := f(s) = \pi f_1(s) + (1 - \pi) f_0(s),$$

where $\pi = \Pr(C = 1)$.

**Definition A.3.** The population-level MLE loss is defined as:

$$\text{MLE}(P_0, P_1) = \mathbb{E}[\textbf{BCELoss}(\mathcal{L}(s; t, \gamma), c)] = \frac{1}{2} \mathbb{E}_{P_1}\big[\log(1 + e^{-\gamma(s-t)})\big] + \frac{1}{2} \mathbb{E}_{P_0}\big[\log(1 + e^{\gamma(s-t)})\big].$$

**Assumption A.4** (Balanced distribution assumption). We further assume that $\pi = \Pr(C = 1) = \Pr(C = 0) = 0.5$,

**Assumption A.5** (Equal-variance location model; correct specification). For each method $m \in \{\text{vcache}, \text{MVR-cache}\}$, we assume that the similarity score conditioned on the label $c$ (which is equal to 0 or 1), $s(x) \mid (C = c)$, follows Gaussian distribution:

$$s(x) \mid (C = 1) \sim P_1 = \mathcal{N}(\mu_1, \sigma^2), \qquad s(x) \mid (C = 0) \sim P_0 = \mathcal{N}(\mu_0, \sigma^2),$$

where $s_m(x)$ denotes the similarity score between a prompt $x$ and its nearest neighbor from the cache store using either vcache or MVR-cache, $\mu_c$ is the mean of the Gaussian distribution, dependent on the method $m$ and the annotation label, $c$. We further assume an identical standard deviation of these normal distributions across different $c$ and $m$.

Hence, $f_c$ ($c = 0$ or 1) defined in Definition A.2 denotes the density function of a Gaussian distribution, i.e.:

$$f_c(s) = f(s|C = c) = \frac{1}{\sqrt{2\pi}\sigma} \exp(-\frac{(s - \mu_c)^2}{2\sigma^2}) \tag{11}$$

**Lemma A.6** (Closed-form posterior and logistic parameters). *Under Assumption A.5, the correctness probability defined in Equation* (2), $\eta(s) := \Pr(C = 1 \mid S = s)$ *is rewritten as follows:*

$$\eta(s) = \frac{1}{1 + \exp\{-\gamma(s - t)\}}, \tag{12}$$

*with*

$$\gamma = \frac{\mu_1 - \mu_0}{\sigma^2}, \qquad t = \frac{\mu_1 + \mu_0}{2}. \tag{13}$$

*Proof.* By Bayes' rule,

$$\frac{\eta(s)}{1 - \eta(s)} = \frac{\Pr(C = 1 \mid s)}{\Pr(C = 0 \mid s)} = \frac{\Pr(s \mid C = 1)\Pr(C = 1)}{\Pr(s \mid C = 0)\Pr(C = 0)} = \frac{f_1(s)}{f_0(s)}.$$

Taking logs and substituting $f_c$ with Equation (11), yielding

$$
\begin{aligned}
\log \frac{\eta(s)}{1 - \eta(s)} &= \log \frac{f_1(s)}{f_0(s)} \\
&= -\frac{(s - \mu_1)^2 - (s - \mu_0)^2}{2\sigma^2} \\
&= -\frac{(s^2 - 2s\mu_1 + \mu_1^2) - (s^2 - 2s\mu_0 + \mu_0^2)}{2\sigma^2} \\
&= \frac{\mu_1 - \mu_0}{\sigma^2} s - \frac{\mu_1^2 - \mu_0^2}{2\sigma^2} \\
&= \frac{\mu_1 - \mu_0}{\sigma^2} \left( s - \frac{\mu_1 + \mu_0}{2} \right).
\end{aligned}
\tag{14}
$$

Furthermore, according to the definition of $\eta(s)$ in Equation (12), $\log \frac{\eta(s)}{1-\eta(s)}$ is further derived as:

$$
\log \frac{\eta(s)}{1 - \eta(s)} = \log \frac{1}{\exp\{-\gamma(s - t)\}} = \gamma(s - t)
$$

By comparing the above formula against Equation (14), we can obtain $\gamma := (\mu_1 - \mu_0)/\sigma^2$ and thus $t = \frac{\mu_1 + \mu_0}{2}$. $\square$

### A.2. Proof of Theorem 3.3

**Theorem A.7.** *Let $\Delta := \mu_1 - \mu_0$. Under the above assumptions, the population-level MLE loss, $MLE(P_0, P_1)$ satisfies:*

1. *$MLE(P_0, P_1)$ depends on $(P_0, P_1)$ only through $\Delta$.*

2. *$MLE(P_0, P_1)$ is strictly decreasing in $\Delta$ for $0 < \Delta < 1$.*

3. *Consequently, the global minimum of $MLE(P_0, P_1)$ over all feasible $(P_0, P_1)$ is attained at*

$$
\mu_1 = 1, \quad \mu_0 = 0.
$$

*Proof.* **Step 1: Explicit form of the risk** Since the class prior is balanced, the population risk can be written as

$$
\text{MLE}(P_0, P_1) = \frac{1}{2} \mathbb{E}_{P_1} \left[ \log(1 + e^{-\gamma(s-t)}) \right] + \frac{1}{2} \mathbb{E}_{P_0} \left[ \log(1 + e^{\gamma(s-t)}) \right].
$$

Substituting $\gamma = \Delta/\sigma^2$ and $t = (\mu_1 + \mu_0)/2$, we obtain

$$
\text{MLE}(P_0, P_1) = \frac{1}{2} \mathbb{E}_{P_1} \left[ \log\left( 1 + \exp\left( -\frac{\Delta}{\sigma^2} \left( s - \frac{\mu_1 + \mu_0}{2} \right) \right) \right) \right] + \frac{1}{2} \mathbb{E}_{P_0} \left[ \log\left( 1 + \exp\left( \frac{\Delta}{\sigma^2} \left( s - \frac{\mu_1 + \mu_0}{2} \right) \right) \right) \right].
$$

**Step 2: Translation invariance.** Define the centered variable

$$
u := s - \frac{\mu_1 + \mu_0}{2}.
$$

Then

$$
\mathbb{E}[u \mid y = 1] = \frac{\Delta}{2}, \qquad \mathbb{E}[u \mid y = 0] = -\frac{\Delta}{2}.
$$

Under equal variance and bounded support, the distributions of $u \mid y = 1$ and $u \mid y = 0$ are translations of each other. Therefore, the joint law of $u$ depends on $(P_0, P_1)$ only through $\Delta$. Hence,

$$
\text{MLE}(P_0, P_1) = \text{MLE}(\Delta),
$$

which proves statement (1).

**Step 3: Derivative with respect to $\Delta$.** Differentiating under the expectation (justified by bounded support and smoothness),

$$\frac{d\text{MLE}}{d\Delta} = -\frac{1}{2\sigma^2}\mathbb{E}_{P_1}\left[\frac{u}{1+\exp\left(\frac{\Delta}{\sigma^2}u\right)}\right] + \frac{1}{2\sigma^2}\mathbb{E}_{P_0}\left[\frac{u}{1+\exp\left(-\frac{\Delta}{\sigma^2}u\right)}\right].$$

Using the symmetry $u \mid y = 0 \overset{d}{=} -u \mid y = 1$, this simplifies to

$$\frac{d\text{MLE}}{d\Delta} = -\frac{1}{\sigma^2}\mathbb{E}_{P_1}\left[\frac{u}{1+\exp\left(\frac{\Delta}{\sigma^2}u\right)}\right].$$

**Step 4: Sign of the derivative.** Under $P_1$, the random variable $u$ has strictly positive mean $\Delta/2$ and bounded support. The function

$$g(u) = \frac{u}{1+\exp\left(\frac{\Delta}{\sigma^2}u\right)}$$

has strictly positive expectation for $\Delta > 0$. Therefore,

$$\frac{d\text{MLE}}{d\Delta} < 0 \quad \text{for all } \Delta \in (0,1),$$

which proves statement (2).

**Step 5: Optimality under bounded scores.** Since $s \in [0,1]$, we have

$$0 \le \mu_0 \le \mu_1 \le 1 \quad \Rightarrow \quad 0 \le \Delta \le 1.$$

Because $\text{MLE}(\Delta)$ is strictly decreasing on $[0,1]$, its minimum is attained at $\Delta = 1$, i.e.,

$$\mu_1 = 1, \quad \mu_0 = 0.$$

This proves statement (3). $\qquad\square$

Theorem A.7 indicates that adapting the similarity score $s$ by minimizing the population-level MLE loss can push $\mu_1$ and $\mu_0$ to 1 and 0, respectively, thus maximizing the gap between these two variables, which is equivalent to maximizing $\gamma$.

Plus, the following lemma suggests the stability of the value of $t$ throughout training:

**Lemma A.8** (Midpoint Stability). *Under the above assumptions, the value of $t$ is invariant under the gradient flow of the following logistic loss:*

$$\ell(s,y) = -y\log g(s) - (1-y)\log(1-g(s)),$$

*in which,*

$$g(s) = \sigma(\gamma(s-t))$$

*If each score $s$ is updated via*

$$\dot{s} = -\frac{\partial\ell}{\partial s} = -\gamma(g(s)-y),$$

*then*

$$\dot{t} = 0,$$

*Proof.* The logistic loss for a sample $(s,y)$ is

$$\ell(s,y) = -y\log g(s) - (1-y)\log(1-g(s)),$$

with gradient

$$\frac{\partial\ell}{\partial s} = \gamma(g(s)-y).$$

The dynamics of the class means are:

$$\dot{\mu}_1 = \mathbb{E}[\dot{s} \mid y = 1] = -\gamma \, \mathbb{E}[g(s) - 1 \mid y = 1] = \gamma \, \mathbb{E}[1 - g(s) \mid y = 1],$$

$$\dot{\mu}_0 = \mathbb{E}[\dot{s} \mid y = 0] = -\gamma \, \mathbb{E}[g(s) - 0 \mid y = 0] = -\gamma \, \mathbb{E}[g(s) \mid y = 0].$$

Thus the midpoint evolves according to

$$\dot{t} = \frac{\dot{\mu}_1 + \dot{\mu}_0}{2} = \frac{\gamma}{2} \left( \mathbb{E}[1 - g(s) \mid y = 1] - \mathbb{E}[g(s) \mid y = 0] \right).$$

Since $f_1$ and $f_0$ are the density function of two normal distributions of the same variance while $t$ is the mid point of the means of these two normal distributions, then $f_1(t + \delta) = f_0(t - \delta)$. Plus, based on the definition of the function $g(\cdot)$, $g(t + \delta) = 1 - g(t - \delta)$ holds. Hence, we have

$$\mathbb{E}[1 - g(s) \mid y = 1] = \int (1 - g(t + \delta)) \, f_1(t + \delta) \, d\delta = \int g(t - \delta) \, f_0(t - \delta) \, d\delta = \mathbb{E}[g(s) \mid y = 0].$$

Hence,

$$\dot{m} = \frac{\gamma}{2} \left( \mathbb{E}[g(s) \mid y = 0] - \mathbb{E}[g(s) \mid y = 0] \right) = 0.$$

At the same time, the inter-class separation evolves as

$$\frac{d}{d\tau}(\mu_1 - \mu_0) = \dot{\mu}_1 - \dot{\mu}_0 = \gamma \left( \mathbb{E}[1 - g(s) \mid y = 1] + \mathbb{E}[g(s) \mid y = 0] \right) > 0,$$

which is strictly positive as long as $\mu_1 < 1$ and $\mu_0 > 0$. Therefore, $t$ is stable while the separation grows monotonically. $\square$

The above analysis suggests that minimizing the MLE loss can maximize $\mu_1$, minimize $\mu_0$, maximize $\gamma$, and maintain the value of $t$. Hence, for those samples with positive labels, $\mathcal{L}(s; t, \gamma)$ increases, and $\tau$ defined in Equation (4) decreases, thus decreasing the exploration probability and increasing the cache hit rate while the same error bound $\tau$ remains fixed.

### A.3. Proof of Lemma 3.4

In the case of the imbalanced class distribution, we can then consider the following class-balanced version of the MLE loss:

$$\sum_i \left[ \frac{1}{\pi} \cdot \sum_{y=1} \log(1 + e^{-\gamma(s-t)}) + \frac{1}{1 - \pi} \sum_{y=0} \log(1 + e^{\gamma(s-t)}) \right],$$

in which the class prior $\pi = \Pr(c = 1)$ could be estimated by the ratio of the positive samples in the training set.

The corresponding population-level MLE loss is thus derived as:

$$\text{MLE}(P_0, P_1) = \mathbb{E}[\textbf{BCELoss}(\mathcal{L}(s; t, \gamma), c)] = \Pr(c = 1) \, \mathbb{E}_{P_1} \left[ \frac{1}{\pi} \cdot \log(1 + e^{-\gamma(s-t)}) \right] + \Pr(c = 0) \, \mathbb{E}_{P_0} \left[ \frac{1}{1 - \pi} \cdot \log(1 + e^{\gamma(s-t)}) \right]$$

$$= \mathbb{E}_{P_1} \left[ \log(1 + e^{-\gamma(s-t)}) \right] + \mathbb{E}_{P_0} \left[ \log(1 + e^{\gamma(s-t)}) \right]$$

This indicates that the population-level MLE loss remains the same. Hence, we can follow the same proof of Theorem 3.3 to prove Lemma 3.4.

## B. Additional experimental setup

### B.1. Additional details for baseline methods

We provide the details of adapting ColBert and POQD to vCache as follows, which both use the symmetric MaxSim score as the similarity measure as MVR-cache:

- **ColBert**(Khattab & Zaharia, 2020): As a pioneering MVR method, it decomposes text into individual tokens and embeds each separately. We adapt it for vCache by encoding tokens from cached and new prompts to produce sequences of embeddings.
- **POQD**(Liu et al., 2025): A state-of-the-art MVR segmentation method that fine-tunes the prompt prefix for a general-purpose LLM to segment queries. For documents in storage, POQD applies heuristic decomposition (e.g., splitting into sentences). Accordingly, we segment cached prompts by sentences, while segmenting incoming prompts using POQD, where fine-tuning is performed using the same training set as MVR-cache.

### B.2. Configurations for MVR-cache

Throughout the experiments, the candidate split positions $\mathcal{P}_x$ are defined as the indices of all punctuation marks in the prompt $x$. To enable efficient retrieval during inference, we use a two-stage retrieval pipeline. We first construct an HNSW index (Malkov & Yashunin, 2016) on the single-vector representations of cached prompts and retrieve the Top-20 nearest-neighbor candidates. We then rerank these candidates using the learned segmentation-aware MaxSim score $\text{SMaxSim}_\Theta$ to select the final nearest neighbor. Thus, MVR-cache uses single-vector retrieval only as a coarse candidate generator and does not require the single-vector Top-1 result to be correct, as long as the correct neighbor is included in the Top-20 candidates.

We validate this design on PromptBench: this two-stage retrieval pipeline achieves a nearest-neighbor recall of 0.179, which is close to 0.183 from a full MVR scan over the entire cache, while avoiding the substantially higher overhead of exhaustive multi-vector retrieval. This small gap indicates that Top-20 single-vector retrieval preserves most relevant candidates, and the MVR reranker can correct many single-vector Top-1 errors. While recent methods such as PLAID (Santhanam et al., 2022) enable efficient retrieval directly over multi-vector data, our preliminary tests show that they introduce substantial computational overhead and do not meet our real-time inference requirements.

## C. Additional Discussion

### C.1. Additional Related Work on Query Reformulation

A related line of retrieval work improves matching quality through query expansion (QE) and pseudo-relevance feedback (PRF), which reformulate the original query using related or feedback-derived terms (Azad & Deepak, 2017; Tu et al., 2025). These methods offer a different latency–quality trade-off from embedding-based approaches: they can improve recall with relatively low overhead, but primarily rely on lexical reformulation rather than learned semantic matching. Recent work further extends this direction with learning-based query reformulation, including reinforcement-learning methods (Wang et al., 2020) and neural rewriting for conversational search (Qian & Dou, 2022). Similar ideas also appear in semantic and approximate caching systems (Ren et al., 2003; Bergman et al., 2025), including ROSE (Luo et al., 2022), which improves cache robustness to misspellings and approximate matches through rewriting.

Our method is related in spirit to query reformulation, but differs in both objective and mechanism. Instead of generating rewritten queries, MVR-cache learns a variable-number segmentation of each prompt and matches the resulting segments to cached prompts using a learned segmentation-aware similarity function. This allows the cache to compare prompts at a finer semantic granularity without replacing the original prompt with a rewritten textual form.

### C.2. Limitations and Future Work

MVR-cache currently segments prompts as linear text sequences. This design applies directly to multi-turn conversations by concatenating the system prompt, dialogue history, and current user query before segmentation. Extending the method to multimodal inputs requires new decomposition mechanisms, such as region-level units for images or temporal units for audio, and is left for future work.

## D. Additional experimental results

We include the experimental results with varied values of $\delta$ in this section.

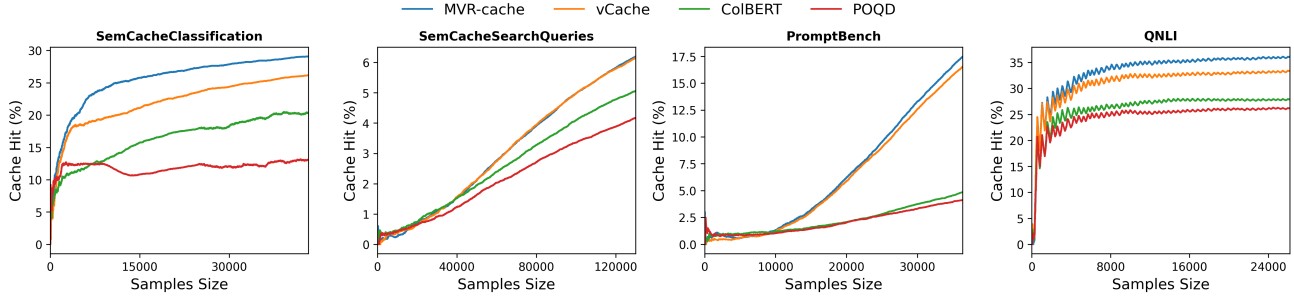

*Figure 8.* The cumulative cache hit rate VS increasing number of incoming prompts with $\delta = 0.02$

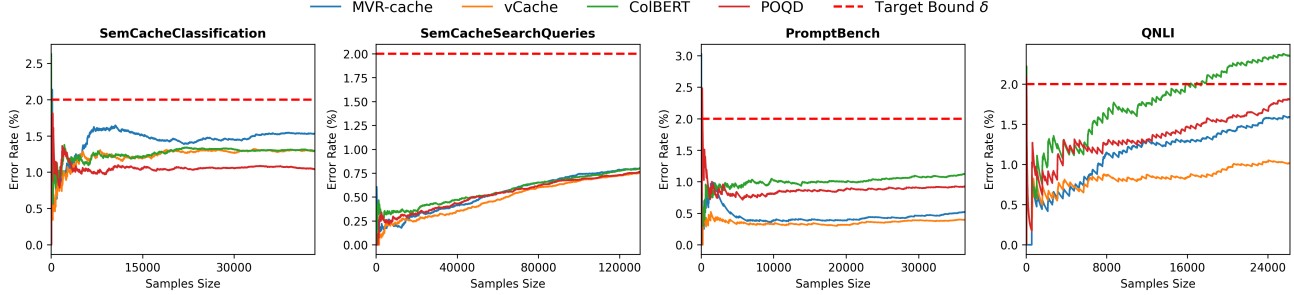

*Figure 9.* The cumulative error rate VS increasing number of incoming prompts with $\delta = 0.02$

### D.1. Cache hit and error rate under varied $\delta$

We further report the cache hit and error rate by varying $\delta$ among $\{0.015, 0.02, 0.03, 0.05, 0.07, 0.08\}$. These results are included in Figure 8-19, which suggest that consistent performance gains of MVR-cache in comparison to the baseline methods.

### D.2. Online overhead breakdown

Table 2 reports the average per-prompt latency breakdown. This table complements Table 1: Table 1 reports cumulative end-to-end latency, while Table 2 decomposes the per-prompt cost into segmentation, embedding, retrieval/reranking, and one LLM call. All values are reported in milliseconds.

As shown in Table 2, the online non-LLM overhead of MVR-cache is small compared with the latency of a single LLM call. The main additional cost comes from the lightweight segmentation step, while retrieval/reranking remains below 0.4 ms per prompt. Therefore, even for infrequent queries where this overhead cannot be amortized over many cache hits, LLM inference remains the dominant cost.

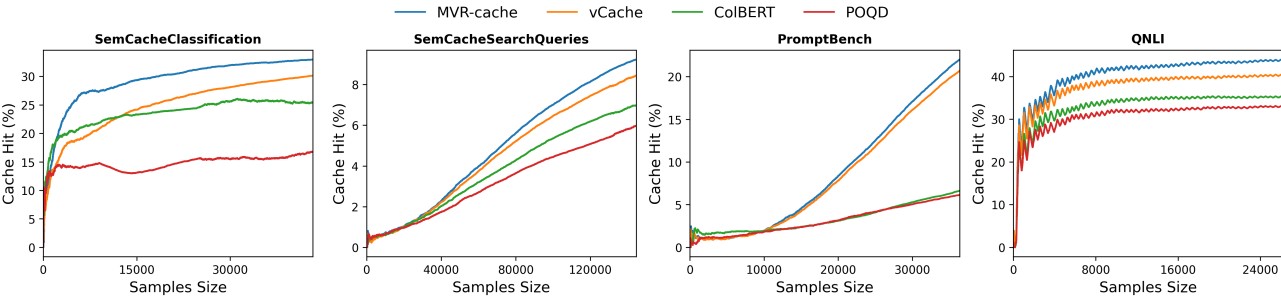

*Figure 10.* The cumulative cache hit rate VS increasing number of incoming prompts under the *cache-on-miss* protocol with $\delta = 0.03$

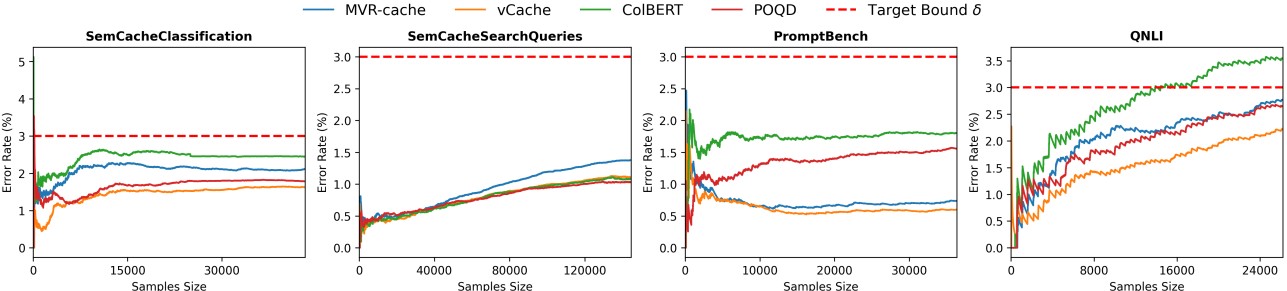

*Figure 11.* The cumulative error rate VS increasing number of incoming prompts under the *cache-on-miss* protocol with $\delta = 0.03$

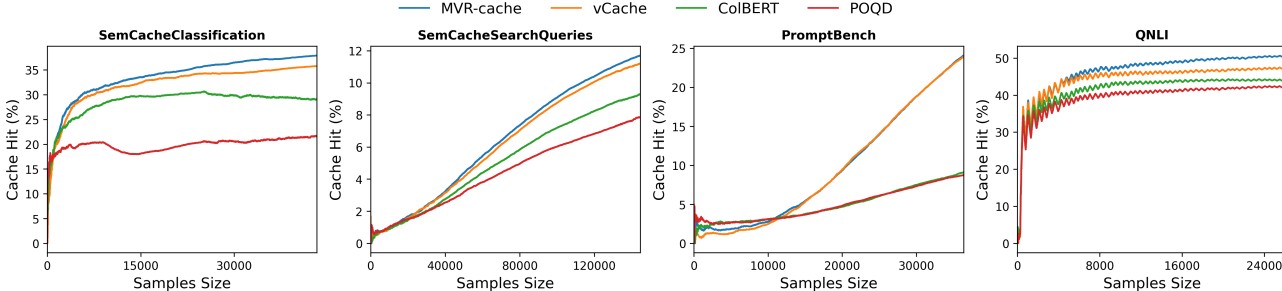

*Figure 12.* The cumulative cache hit rate VS increasing number of incoming prompts under the *cache-on-miss* protocol with $\delta = 0.05$

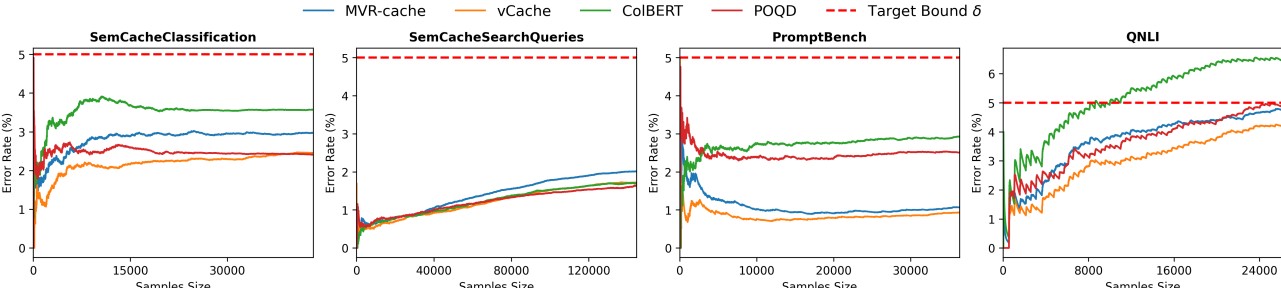

*Figure 13.* The cumulative error rate VS increasing number of incoming prompts under the *cache-on-miss* protocol with $\delta = 0.05$

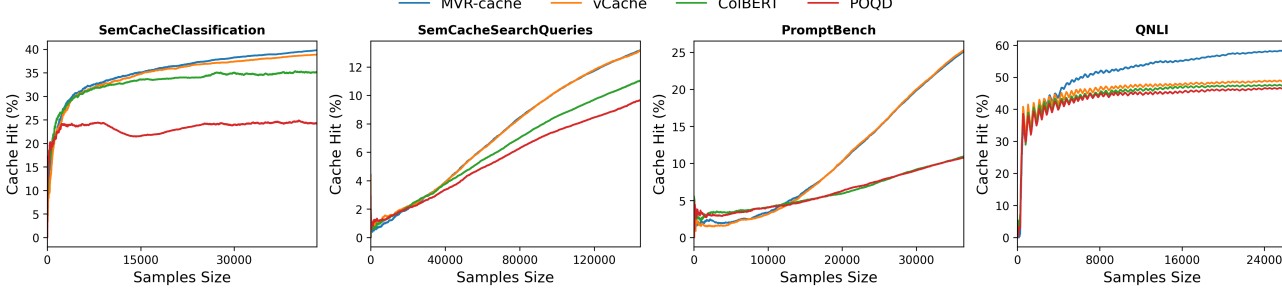

*Figure 14.* The cumulative cache hit rate VS increasing number of incoming prompts under the *cache-on-miss* protocol with $\delta = 0.07$

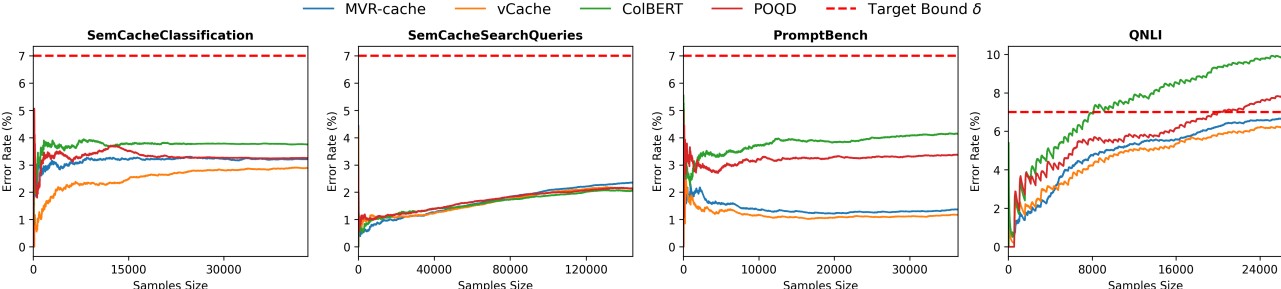

*Figure 15.* The cumulative error rate VS increasing number of incoming prompts under the *cache-on-miss* protocol with $\delta = 0.07$

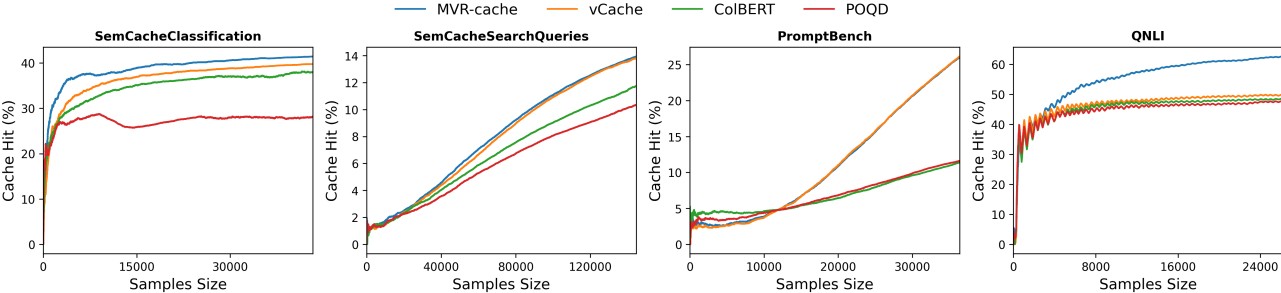

*Figure 16.* The cumulative cache hit rate VS increasing number of incoming prompts under the *cache-on-miss* protocol with $\delta = 0.08$

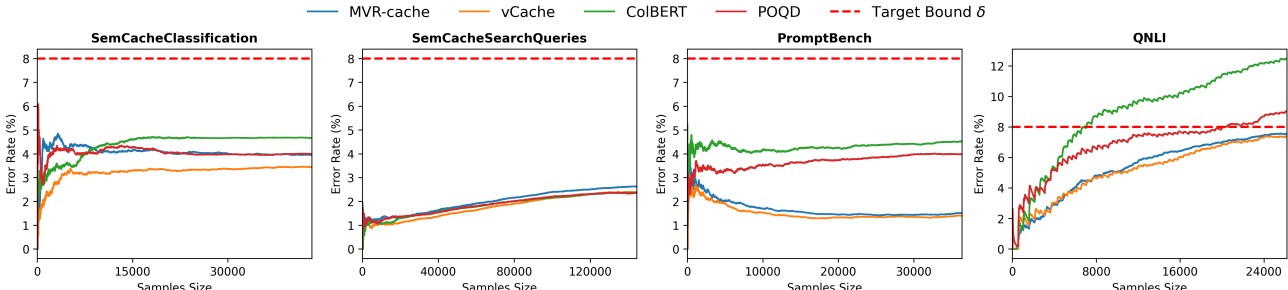

*Figure 17.* The cumulative error rate VS increasing number of incoming prompts under the *cache-on-miss* protocol with $\delta = 0.08$

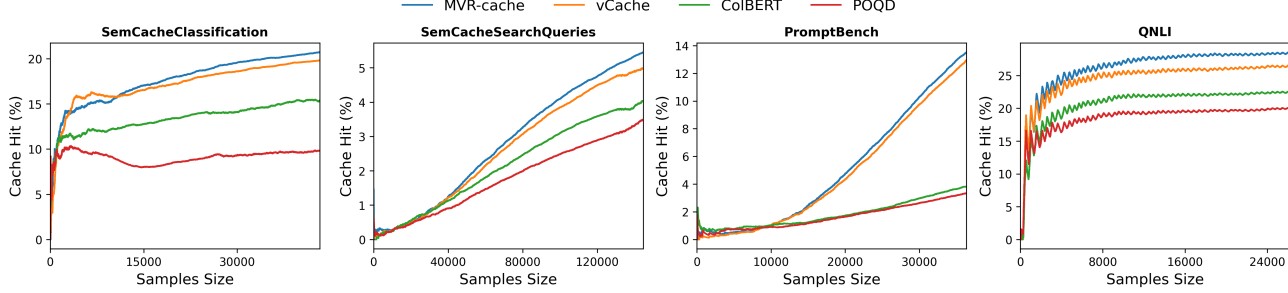

*Figure 18.* The cumulative cache hit rate VS increasing number of incoming prompts under the *cache-on-miss* protocol with $\delta = 0.015$

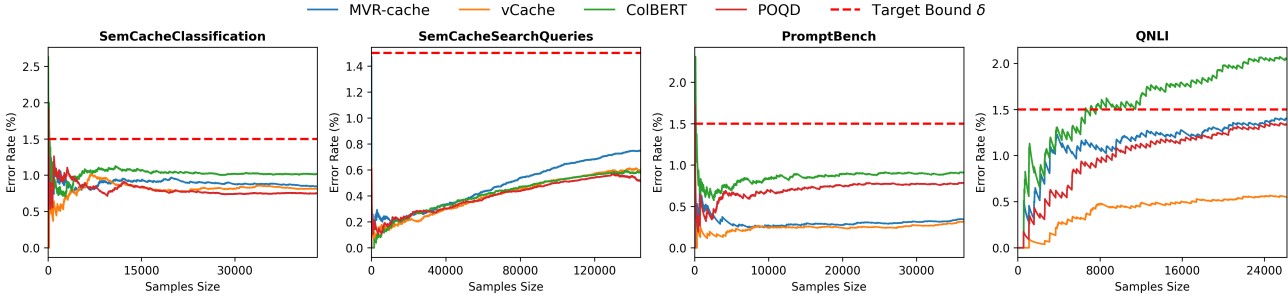

*Figure 19.* The cumulative error rate VS increasing number of incoming prompts under the *cache-on-miss* protocol with $\delta = 0.015$

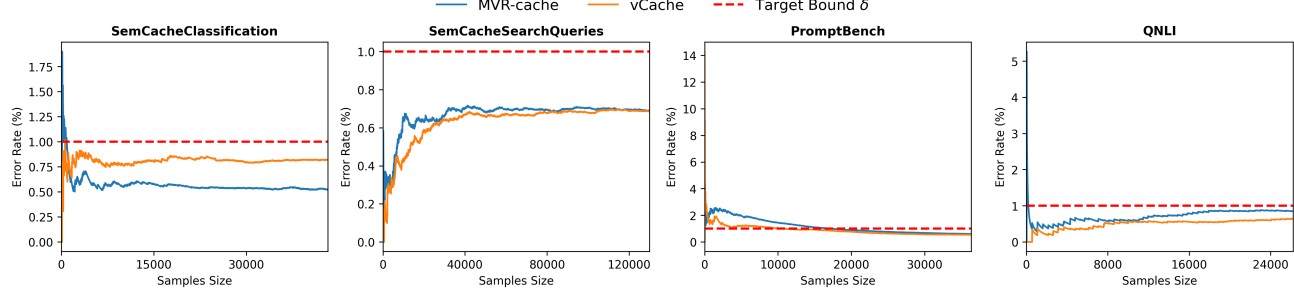

*Figure 20.* Cumulative error rate e VS increasing number of incoming prompts under the *always-cache* protocol with $\delta = 0.01$.

### D.3. Ablation studies

We further perform ablation studies for MVR-cache using the promptbench dataset with the error bound $\delta = 0.01$. First of all, in Figure 21, we compare the effect of MVR-cache using varied embedding models including the default BGE model, GTE Large model (Wang et al., 2022) and E5-large (Wang et al., 2022), which suggest that there is almost no performance difference for MVR-cache between different embedding models.

In addition, we also vary the number of training samples for training the segmentation model. As Figure 22 shows, regardless of the training set sizes, MVR-cache ends up with almost the same performance. This thus indicates that a training set with 3K training samples is sufficient to obtain a reasonable segmentation model.

In the main experiments, we use punctuation marks as candidate split positions. To test whether this design choice limits the segmentation policy, we compare it with three alternative candidate sets on PromptBench while keeping the rest of the MVR-cache pipeline fixed: keyword-level boundaries, token-level boundaries, and sentence-level boundaries. Keyword-level boundaries include punctuation marks and selected keywords such as "and" and "or"; token-level boundaries

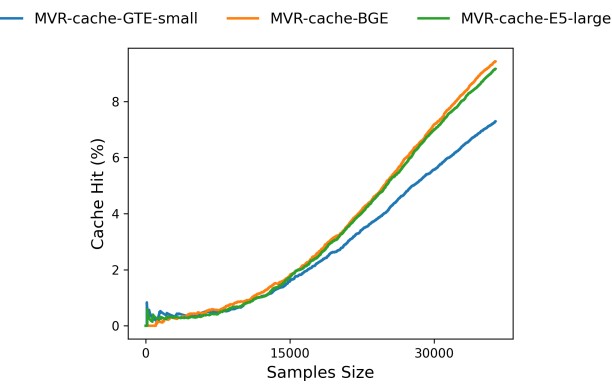

*Figure 21.* Ablation on embedding models

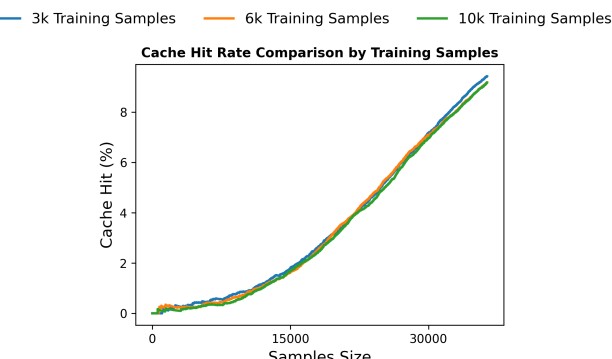

*Figure 22.* Ablation on the training set size

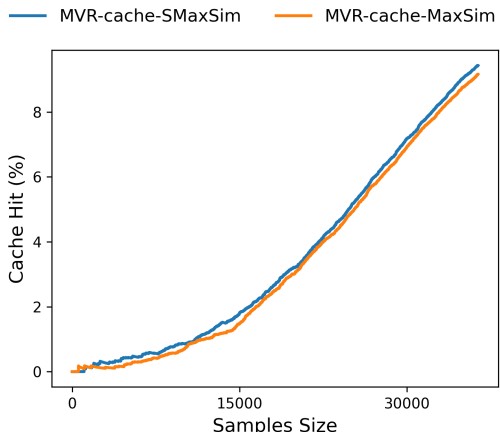

*Figure 23.* Comparison of using the symmetric MaxSim score and the vanilla MaxSim score

*Table 2.* Average per-prompt latency breakdown. All values are in milliseconds. The non-LLM total excludes the LLM call.

| Dataset | Method | Seg. | Emb. | Ret./rerank | Non-LLM total | LLM call |
|---|---|---|---|---|---|---|
| SemCacheCls. | MVR-cache | 23.00 | 32.00 | 0.35 | 55.35 | 1234.60 |
| | vCache | – | 25.00 | 0.20 | 25.20 | 1234.60 |
| SemCacheSQ | MVR-cache | 28.00 | 32.00 | 0.35 | 60.35 | 3004.20 |
| | vCache | – | 23.00 | 0.20 | 23.20 | 3004.20 |
| PromptBench | MVR-cache | 22.00 | 32.00 | 0.35 | 54.35 | 3352.00 |
| | vCache | – | 25.00 | 0.30 | 25.30 | 3352.00 |
| QNLI | MVR-cache | 23.00 | 32.00 | 0.35 | 55.35 | 4273.00 |
| | vCache | – | 20.00 | 0.30 | 20.30 | 4273.00 |

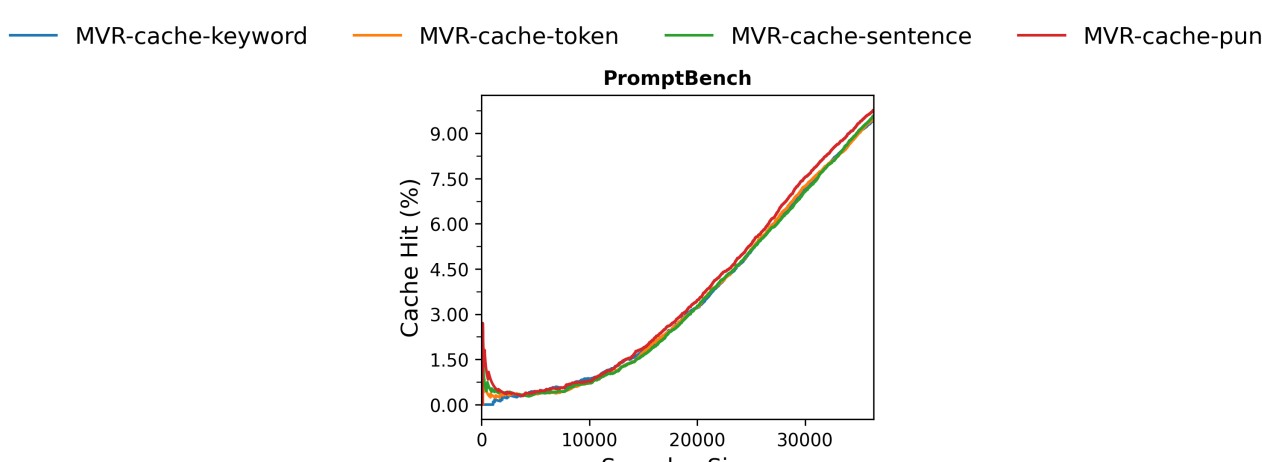

*Figure 24.* Sensitivity analysis of candidate split positions on PromptBench. We report the cumulative cache hit rate as the number of incoming prompts increases.

include punctuation marks and spaces; sentence-level boundaries include punctuation marks excluding commas; and punctuation-level boundaries correspond to our default setting.

Figure 24 shows the cumulative cache hit rate as the number of incoming prompts increases. The curves are highly similar across all candidate sets, indicating that giving the policy a larger set of possible split positions does not lead to a meaningful improvement. In particular, punctuation-level splitting remains competitive with, and slightly better than, the larger candidate sets in the final cache hit rate. This suggests that punctuation marks already provide a compact and effective search space for prompt segmentation in our setting.

We also report the number of segments selected by the learned policy in Table 3. The average segment count is close to one for short search queries and increases for longer question-answering prompts. This shows that the policy does not use a fixed segmentation length, but adapts the number of segments to the input structure.

In the end, we also replace the symmetric MaxSim score, i.e., SMaxSim, used in MVR-cache with the vanilla MaxSim score, which is unidirectional. The results are reported in Figure 23, which reveals a slight performance gain if the symmetric MaxSim score is used.

### D.4. Qualitative Example on Long-Context Prompts

We further examine whether MVR-cache remains effective for very long prompts. Long-context inputs are challenging for semantic caching because a single-vector representation can dilute local semantic details. A long prompt may contain many repeated or related sub-questions, and the information that determines response equivalence may appear only in specific parts of the input. As a result, single-vector retrieval may retrieve a broadly related but response-inequivalent nearest neighbor.

*Table 3.* Statistics of the number of segments selected by MVR-cache.

| DATASET | MIN | MAX | MEAN |
|---|---|---|---|
| SEMCACHESEARCHQUERIES | 1 | 4 | 1.01 |
| SEMCACHECLASSIFICATION | 1 | 93 | 2.64 |
| QNLI | 1 | 149 | 5.31 |
| PROMPTBENCH | 1 | 262 | 7.67 |

*Table 4.* A qualitative long-context example. The incoming prompt is about the CASA 1000 electricity transmission project. MVR-cache retrieves a response-equivalent CASA 1000 prompt and produces a cache hit, while vCache retrieves a response-inequivalent prompt about solar hot water systems and misses the cache.

| **Incoming prompt excerpt** | | |
|---|---|---|
| "Describe the project named CASA 1000. What is the intended purpose of the CASA 1000 project? . . . CASA 1000 will transmit 1000 MW of surplus electricity from Tajikistan to Pakistan with power transit through Afghanistan. . . . Further clarify the role and involvement of Tajikistan, Pakistan, and Afghanistan. . . . Describe potential risks, timeline, budget, revenue sharing, and regional impacts." | | |
| Method | Retrieved nearest-neighbor prompt excerpt | Result |
| vCache | "Explain in detail the capacity of a solar hot water system . . . including global installations as of 2007, approximately 154 thermal gigawatt (GWth). . . . What does this capacity represent in terms of the number of homes, hospitals, or businesses that can be supported? . . . Describe the geographical distribution of these installations and the average system size." | Miss |
| MVR-cache | "The CASA 1000 project promotes regional energy security through regional cooperation. . . . Explain how regional cooperation on energy projects like CASA 1000 enhances regional energy security. . . . Describe the planned project objectives, participating countries, infrastructure deployment, risks, and regional economic implications." | Hit |

Table 4 shows a representative example. The incoming prompt contains around 10K tokens and consists of many sub-questions about the CASA 1000 electricity transmission project. To make the example readable, we show only representative excerpts from the incoming prompt and the retrieved nearest-neighbor prompts, omitting repeated sub-questions with ellipses.

The difference comes from how the two methods represent the long prompt. vCache compresses the entire prompt into one embedding. In this example, the retrieved prompt is broadly energy-related, but it concerns solar hot water capacity rather than the CASA 1000 project. Since the two prompts require different LLM responses, vCache misses the cache.

In contrast, MVR-cache decomposes the long prompt into multiple segments and compares them with cached prompts using SMaxSim$_\Theta$. The learned segments preserve specific local semantics, such as the project goal, the participating countries, implementation risks, and regional impacts. These segment-level matches allow MVR-cache to retrieve a nearest neighbor from the same response-equivalence set and produce a correct cache hit. This example illustrates why variable-length segmentation is useful for long-context prompts: the relevant matching evidence may be distributed across several local spans and can be obscured by a single global representation.

### D.5. Empirically validate Assumption 3.1

Recall that our theoretical results rely on Assumption 3.1, which states that the class-conditional distributions of the learned segmentation-aware similarity scores approximately follow normal distributions. To empirically validate this assumption, we evaluate the learned SMaxSim$_\Theta$ scores on all four datasets. Specifically, for each dataset, we randomly sample prompts and compute their similarity scores against their retrieved nearest-neighbor prompts, then group the scores according to their corresponding labels. As shown in Figures 25 - 28, the resulting class-conditional score distributions are approximately normal across all datasets, providing empirical support for Assumption 3.1. For the additional assumption used in Theorem 3.3, i.e., Assumption 3.2, we acknowledge that it may be strong in practice. To address this, Lemma 3.4 shows that under Assumption 3.1 alone, minimizing the class-rebalanced objective still improves the cache hit rate under the same error bound. Therefore, since Assumption 3.1 is empirically supported across all four datasets, the theoretical implication of Lemma 3.4 remains meaningful.

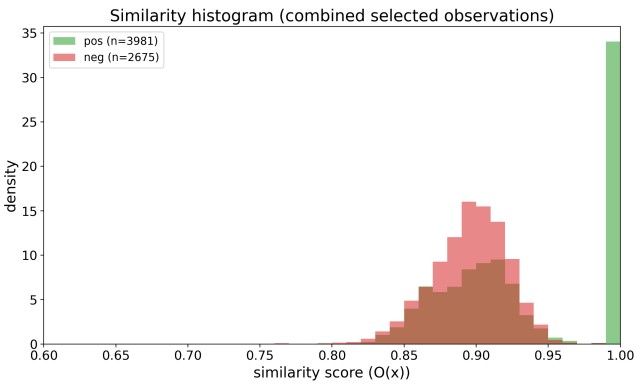

*Figure 25.* Empirically validate Assumption 3.1 on SemCacheSearchClassification dataset

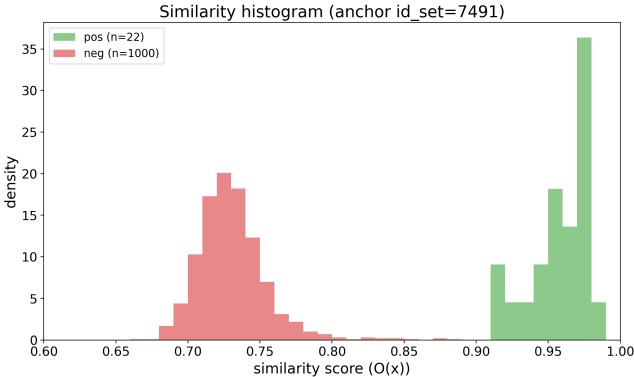

*Figure 26.* Empirically validate Assumption 3.1 on SemCacheSearchQueries dataset

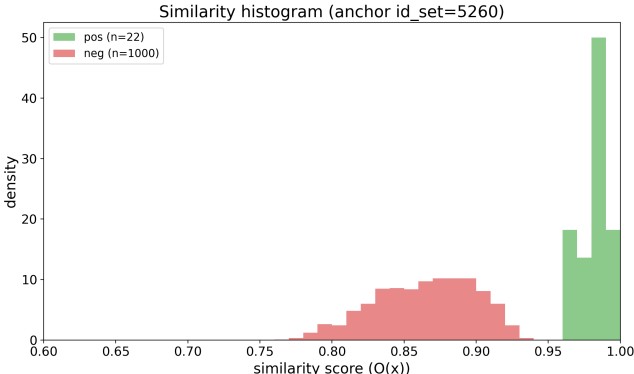

*Figure 27.* Empirically validate Assumption 3.1 on PromptBench dataset

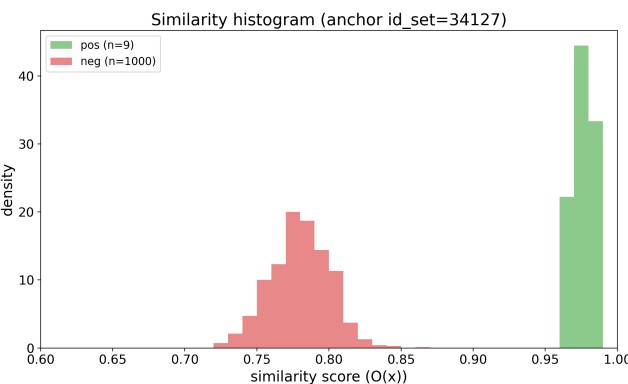

*Figure 28.* Empirically validate Assumption 3.1 on QNLI dataset

