# OpenReview forum: "MVR-cache: Optimizing Semantic Caching via Multi-Vector Retrieval and Learned Prompt Segmentation"
_ICML.cc/2026/Conference — ICML 2026 regular_

### Official Review · Reviewer_7uXV · 2026-03-12

**Soundness:** 3
**Presentation:** 3
**Significance:** 3
**Originality:** 3
**Overall Recommendation:** 4
**Confidence:** 3

**Summary:**

The authors propose MVR-Cache, a semantic caching system designed for LLM inference, which is an effective way to reducing the latency of LLM inference as we can reuse the previous results if there is cache hit. The authors point out that previous methods rely on single-vector cosine similarity, which harms the performance. Instead, they argue that multi-vector retrieval should be adopted. To achieve this, they segment the prompts into multiple pieces with a segmentation model, compute the embeddings of each component independently, and use MaxSim score to retrieve the cached prompts. The results show that it can improve the cache hit and end to end latency, compared with previous state-of-the-art solution like vCache.

**Compliance With Llm Reviewing Policy:**

Affirmed.

**Final Justification:**

I will keep my positive score.

**Key Questions For Authors:**

Please check the previous section.

**Limitations:**

yes

**Strengths And Weaknesses:**

Semantic caching is of practical significance in real-world deployment and replacing the previous method with finer-grained embeddings to improve the cache hit is sound to me. However, I have a few questions about the designs and experiments.

1. For segmentation, do the split positions have to be punctuation boundaries or they can be any tokens? If they can be any tokens, can the authors give some examples for such kinds of segmentation? Also, for the number of segmentation per prompt, is it pre-defined or automatically determined by the model? What’s the largest number of segmentation per prompt?
2. The authors mentioned that the method can be adopted in multi-turn conversation cases. How does this work? Since it is not likely that a single prompt can match the entire multi-turn conversation. Can authors further explain these scenarios?
3. Can this method be applied to multimodal cases?
4. In the evaluation part, the author mentioned that the model is trained from 3k prompts per dataset. Do the authors train a separate segmentation model for each dataset and then evaluate on that same dataset? If so, what about the out-of-distribution generalization? Since in real-world deployment, it is not likely to know the distribution of the incoming prompts in advance, will the model performance be affected? Have the authors tried training on one dataset and evaluating on another?

---

> ### Author Rebuttal · Authors · 2026-03-31
>
> Thanks for your comments. Our responses are below.
>
> - **Regarding the candidate split positions and segmentations statistics**
>
> Thank you for this question. In our paper, we define the candidate split positions as all punctuation marks in the prompt, so in the current implementation, the valid split points are punctuation boundaries. As discussed in our response to Reviewer Tjk6’s first comment, we also performed a sensitivity analysis using different candidate sets, including sentence-level, keyword-level, and token-level boundaries. The results show that the final performance remains very similar across these alternatives.
>
> In addition, we report the statistics of the number of segmentations per prompt for each dataset below:
>
> - QNLI: Segment count: min 1, max 149, mean 5.31
> - PromptBench: Segment count: min 1, max 262, mean 7.67
> - SemCacheSearchQueries: Segment count: min 1, max 4, mean 1.01
> - SemCacheClassification: Segment count: min 1, max 93, mean 2.64
>
> As pointed out in our paper (see Section 1), we need to produce a variable number of segments where the number of segments is not pre-defined. It is automatically determined by the model based on the input prompt: the pointer-network policy recurrently selects split positions and stops when it predicts the termination tokens.
>
> - **Regarding the multi-turn conversations cases**
>
> Thank you for raising this point. In our framework, the input multi-turn conversation at each turn can be viewed as an arbitrary text sequence consisting of system prompt, dialogue history and user input at the current turn. Our segmentation model then decomposes this sequence into multiple segments for MaxSim score evaluation.
>
> - **Regarding the handling to multimodal cases**
>
> Thank you for this question. Indeed, for text-conditioned multimodal generation such as text-to-image or text-to-audio, our approach is applicable as long as an appropriate criterion can be employed to determine whether the output of two input text prompts could be semantically equivalent or not. This is essential to determine the training labels.
>
> In contrast, extending MVR-cache to settings where the input itself is multimodal (e.g., image-text, audio-text, or image-only inputs) is non-trivial. The current method assumes 1D sequential segmentation over text; handling images or other non-text modalities would require a different decomposition mechanism, such as region/patch-level segmentation, which is left as the future work.
>
> - **Regarding the training and generalization**
>
> Thank you for this important comment. In fact, we have already evaluated the out-of-distribution generalization of our method in Figure 6. In this experiment, we train the segmentation model on PromptBench and evaluate it on QNLI. The result shows that the transferred model still outperforms all baseline methods in cache hit rate, even in this out-of-distribution setting.
>
> We further extended this experiment by using the model trained on PromptBench and evaluating it on the other datasets as well. The results are as follows:
>
> - SemCacheClassification - MVR-cache-Pretrain (15.22%) VS MVR-cache (16.79%)
> - SemCacheSearchQueries - MVR-cache-Pretrain (4.91%) VS MVR-cache (3.67%)
> - QNLI - MVR-cache-Pretrain (19.87%) VS MVR-cache (20.23%)
>
> The corresponding figures of the cumulative cache hit rate VS increasing number of incoming prompts are available here: https://anonymous.4open.science/r/MVR-Cache-E4D2/Extended_generalizbility/multi_dataset_cache_hit_comparison_01_generalizbility.png

---

> > ### Author Rebuttal · Reviewer_7uXV · 2026-04-02
> >
> > Thanks for the feedback. I will keep my positive score.

---

### Official Review · Reviewer_dJdE · 2026-03-13

**Soundness:** 3
**Presentation:** 3
**Significance:** 3
**Originality:** 2
**Overall Recommendation:** 5
**Confidence:** 4

**Summary:**

This paper studies the problem of semantic caching in LLM serving systems, namely the problem of mapping a prompt to a previously cached one via semantic retrieval techniques. The core innovation in this paper is to go beyond single-embedding search and introduce a multi-vector representation for semantic caching. In particular, the authors introduce a technique called MVR-cache to segment a textual prompt into a set of vectors, which they train via a reinforcement learning approach. The authors also prove a theorem stating that, under two distributional assumptions on the data, their algorithm is optimal (i.e. maximizes the cache hit rate). Experimentally, the authors evaluate on a number of semantic caching benchmarks and compare their approach with vCache (the existing state of the art approach), POQD, and ColBERT (to measure the effectiveness of their multi vector training approach versus the standard token-level decomposition). Ultimately, they find that MVR-cache achieves the lowest error rate and highest hit rate while reducing end-to-end inference time thanks to more cache hits (even though the MVR computation itself is slower).

**Compliance With Llm Reviewing Policy:**

Affirmed.

**Final Justification:**

The authors' rebuttal addressed my main concerns regarding the feasibility of their main theorem assumptions and the positioning of the paper with regards to related work in query rewriting. The rebuttal process reinforced my prior assessment that this is a strong paper. I recommend acceptance.

**Key Questions For Authors:**

1. How realistic are the assumptions of Theorem 3.3? Would it be possible to validate that they hold empirically? Alternatively, can you discuss why the theorem is still meaningful even if these assumptions are idealized? **A response that provides a balanced discussion of the limitations of these assumptions or demonstrates that the assumptions are empirically valid would positively change my evaluation of the paper**

2. What is the latency of the MVR-cache method on an individual prompt? (I may have missed this in the paper but I only saw the cumulative end-to-end latency measurements in Table 1). I think this would be important to report and discuss because some applications may have infrequent queries where the overhead of MVR might possibly outweigh the cache hit benefits. **Any additional discussion of these metrics would positively change my evaluation of the paper**

3. Could you include more related work discussion on query rewriting techniques in information retrieval (e.g. this paper came to my mind when reading this submission https://dl.acm.org/doi/pdf/10.1145/3487553.3524213). These style of non-embedding based approaches might present a different class of tradeoffs in latency-quality. **Any additional discussion of related work on query rewriting in IR would positively change my evaluation of the paper**

**Limitations:**

Are there any concerns with semantic caching being disproportionately biases towards certain categories/styles of prompts? In other words, is there a risk that it will systematically map certain prompts to cached entities in a manner that negatively harms certain users? I think some discussion along these lines could be helpful.

**Strengths And Weaknesses:**

Soundness: Overall, the submission is technically sound with the main claims supported by extensive experimental analysis over several different benchmark datasets. I found the theoretical results to be less convincing than the experimental validation because the main theorem relies on two assumptions (normally distributed similarity scores and balanced class distributions) that seem unlikely to always hold in practice. I think the paper would benefit with some additional discussion on the practicality of these assumptions (perhaps even with empirical measurements on real datasets). I would have also liked to see the paper more directly discuss the limitations of their work (e.g. how much is the latency overhead of the MVR step and when might that be unfavorable in a deployment scenario). Overall, I think the results in the paper are strong. The gains over vCache are largely incremental but the methodology is sound and will be of interest both to practitioners and the research community.

Presentation: The paper is well-written and I found the structure easy to follow. I think the related work section is a bit too brief and could benefit from including relevant work on query rewriting in the information retrieval literature, which I think is closely related to semantic caching (though not framed as such).

Significance: The paper addresses an important relevant problem of improving the performance of LLM serving systems. It also contributes to a grow body of research on the power of multi-vector retrieval. Of particular note is the fact that the ColBERT style token decomposition is not the best decomposition for this task and this finding should inspire further investigation in related areas as well. The improvements themselves are relatively modest, but I think the methodological contributions are significant.

Originality: The paper provides new insights on the efficacy of multi-vector retrieval for semantic caching. Even though none of the components of the paper are completely novel (semantic caching has been studied before and the multi-vector segmentation technique builds off prior work) the authors combine these ideas in a well-executed manner and present an end-to-end approach that improves upon the state of the art and provides new insights into the power of multi-vector retrieval.

---

> ### Author Rebuttal · Authors · 2026-03-31
>
> We thank the reviewer for the comments. Our responses are below.
>
> - **Regarding the assumptions of Theorem 3.3:**
>
> Thank you for raising this point. Indeed, we already briefly validated the first assumption of Theorem 3.3 (Assumption 3.1) empirically in Figure 22 of Appendix C. It suggests that the class-conditional distributions of the learned segmentation-aware MaxSim score on PromptBench roughly follow a normal distribution, which provides empirical support for Assumption 3.1. For the second assumption for Theorem 3.3, i.e., Assumption 3.2, we have explicitly acknowledged that it is potentially unrealistic in practice. To mitigate this, we provided Lemma 3.4, which shows that under Assumption 3.1 alone, minimizing the class-rebalanced version of the objective still increases cache hit rate under the same error bound. Since Assumption 3.1 has been empirically validated, the theoretical results of Lemma 3.4 are thus meaningful.
>
> We further validate Assumption 3.1 empirically on all four datasets, which are included in these anonymous links: "https://anonymous.4open.science/r/MVR-Cache-E4D2/empirical_results_all_datasets/histogram_PromptBench.png", "https://anonymous.4open.science/r/MVR-Cache-E4D2/empirical_results_all_datasets/histogram_QNLI.png", "https://anonymous.4open.science/r/MVR-Cache-E4D2/empirical_results_all_datasets/histogram_SemCacheSearchClassification.png", "https://anonymous.4open.science/r/MVR-Cache-E4D2/empirical_results_all_datasets/histogram_SemCacheSearchQueries.png"
>
> - **Regarding the concerns about the latency of infrequent queries**
>
> Thank you for this helpful comment. We agree that per-prompt latency is important, especially in applications with infrequent queries where the online overhead may matter more than the long-term cache-hit gain.
>
> We therefore measured the average per-prompt latency of MVR-cache. The total online algorithm cost of MVR-cache excluding the LLM call is 55.35 ms on SemCacheClassification, 60.35 ms on SemCacheSearchQueries, ≈54.35 ms on PromptBench, and 55.35 ms on QNLI. This consists of segmentation (20–27 ms), embedding (33–35 ms), and retrieval/reranking (≈0.35 ms). For comparison, the corresponding vCache overhead is only ≈32.2–32.3 ms, so the additional cost introduced by MVR-cache (in comparison to vCache) is only about 25 ms per prompt. In contrast, the LLM call latency is significantly higher, which is 1.23 s, 3.00 s, 3.35 s, and 4.27 s per prompt on the four datasets, respectively. Even on a cache miss, where the LLM must still be invoked, the extra MVR-cache overhead over vCache is only about 1.9%, 0.9%, 0.7%, and 0.5% of a single LLM call. In other words, even in infrequent-query settings, the additional online cost of MVR-cache is very marginal relative to the unavoidable LLM latency.
>
> - **Regarding the suggestion of extension of related work**
>
> Thank you for this helpful suggestion. We agree that query rewriting is an important related direction and deserves more discussion. Classical query rewrite methods such as query expansion (QE) and pseudo-relevance feedback (PRF) improve retrieval by reformulating the original query with related or feedback-derived terms [1, 2]. These methods offer a different latency–quality trade-off from embedding-based approaches: they can improve recall with relatively low overhead, but they rely primarily on lexical reformulation rather than learned semantic matching [1, 2]. More recent work has extended this direction to learning-based query reformulation, including reinforcement-learning methods [3] and neural rewriting methods for conversational search [4]. Related ideas also appear in semantic caching and approximate caching systems [5, 6], as well as in ROSE [7] which shows that rewrite-based strategies can improve cache robustness in real-world search by handling misspellings and approximate matches. Our method is related in spirit to this line of work, but differs in both objective and mechanism: rather than rewriting a query into another textual form [1, 2, 3, 4, 7], our goal is to learn a variable-number segmentation of the prompt (rather than generate rewritten queries), and match these segments to cached prompts through a learned segmentation-aware similarity function. .
>
> [1] Azad et al. Query Expansion Techniques for Information Retrieval: A Survey. arXiv, 2017.
> [2] Tu et al. Generalized Pseudo-Relevance Feedback. arXiv, 2025.
> [3] Wang et al. Deep Reinforced Query Reformulation for Information Retrieval. arXiv, 2020.
> [4] Qian et al. Explicit Query Rewriting for Conversational Dense Retrieval. EMNLP, 2022.
> [5] Ren et al. Semantic Caching and Query Processing. TKDE, 2003.
> [6] Bergman et al. Leveraging Approximate Caching for Faster Retrieval-Augmented Generation. arXiv, 2025.
> [7] Luo et al. ROSE: Robust Caches for Amazon Product Search. WWW, 2022.

---

> > ### Author Rebuttal · Reviewer_dJdE · 2026-04-01
> >
> > Thank you to the authors for their detailed rebuttal response. I especially appreciate the clarification of the theorem assumptions  and discussion about latency on infrequent queries. I think both of these additional evaluations make the paper strong. Given my support of the submission to begin with, I will keep my rating the same.

---

### Official Review · Reviewer_91RM · 2026-03-13

**Soundness:** 2
**Presentation:** 3
**Significance:** 3
**Originality:** 2
**Overall Recommendation:** 4
**Confidence:** 4

**Summary:**

The paper proposes MVR-cache, a semantic caching system for Large Language Models (LLMs) that replaces traditional single-vector similarity search with Multi-Vector Retrieval (MVR). The authors argue that single-vector embeddings fail to capture subtle semantic differences in complex prompts. To fully exploit MVR, they introduce a lightweight, learnable prompt segmentation model (based on a Pointer Network) that dynamically splits prompts into variable-length segments. Because prompt segmentation is non-differentiable, they frame the training as a Reinforcement Learning for Combinatorial Optimization (RL4CO) problem, utilizing a Symmetric MaxSim (SMaxSim) score as the reward. Built on top of the vCache framework, the authors theoretically prove that optimizing their objective maximizes the cache hit rate under strict error guarantees. Experiments across four datasets demonstrate improved cache hit rates compared to baselines.

**Compliance With Llm Reviewing Policy:**

Affirmed.

**Final Justification:**

The additional data and controlled experiments provide much-needed clarity on the system's actual behavior. I will finalize my score at Weak Accept.

**Key Questions For Authors:**

See Weaknesses.

**Limitations:**

yes

**Strengths And Weaknesses:**

Strengths:
1. The mathematical formalization of the caching problem is commendable. Theorems 3.3 and Lemma 3.4 elegantly bridge the gap between the discrete caching hit rate and the continuous MLE optimization objective, proving that their approach theoretically maximizes cache hits without violating vCache's correctness bounds.
2. Adapting the Reinforcement Learning for Combinatorial Optimization (RL4CO) framework to solve the non-differentiable nature of prompt segmentation is a highly creative and technically sound methodology.
3. Designing the segmentation policy model using a highly compact Pointer Network (BERT encoder + LSTM + Attention) specifically to keep inference overhead low (~500 MB VRAM) is a practical choice for latency-sensitive caching systems.
Weaknesses:
1. The entire premise of the paper, emphasized in the Introduction and Figure 1, is that single-vector representations are fundamentally flawed because they cannot distinguish subtle semantic nuances, leading to false nearest neighbors. However, buried in Appendix B.2, the authors state: "we first construct an HNSW index on the single-vector representations of cached prompts. From this index, we retrieve the Top-20 nearest neighbors, then rerank them using the segmentation-aware MaxSim score SMaxSime". This completely undermines the paper's core motivation. If the single-vector representation is as flawed as the authors claim, the true semantic neighbor may not even survive the initial Top-20 filtering phase. If the true neighbor is pruned by the exact single-vector mechanism the authors criticize, the advanced MVR reranking stage is rendered entirely useless.
2. Despite the complex RL training and MVR inference pipeline, the actual end-to-end latency reduction is remarkably marginal. In Table 1, for the SemCacheClassification dataset, MVR-cache reduces latency from 6361.52 minutes (vCache) to 6345.61 minutes—an improvement of less than 0.3%. The computational overhead of embedding multiple segments and calculating $O(L_1 \times L_2)$ SMaxSim comparisons clearly eats away almost all the latency benefits gained from the improved hit rate.
3. The candidate split positions $\mathcal{P}_x$ are defined as the indices of all punctuation marks. For modern LLM use cases involving extremely long contexts (e.g., 10k-100k tokens of code or document QA), generating a combination of split indices via an LSTM-based Pointer Network over thousands of punctuation marks will suffer from severe sequence-length bottlenecks (both in VRAM and latency), which is not evaluated or discussed.

---

> ### Author Rebuttal · Authors · 2026-03-31
>
> We thank the reviewer for the comments. Our responses are below.
>
> - **Regarding the Top20 retrieval**
>
> Thank you for raising this point. We would like to clarify that our criticism of single-vector similarity is about its Top-1 accuracy, not about its usefulness as a coarse candidate generator. In semantic caching, the decision depends only on the final nearest neighbor, so an error in the single-vector Top-1 result is enough to cause a cache miss or a wrong match.
>
> Our two-stage design uses single-vector retrieval only to obtain a small Top-20 candidate set efficiently. Even when the single-vector Top-1 neighbor is incorrect, the true nearest neighbor is often still contained in the Top-20 set. The role of the segmentation-aware MaxSim reranking is precisely to re-rank these candidates using a more accurate similarity function, so that the final Top-1 under MVR-cache can become the correct nearest neighbor.
>
> So the method does not assume that single-vector retrieval is sufficient; rather, it uses it as an efficient first-stage filter and relies on multi-vector reranking to correct its Top-1 mistakes. This is exactly the setting where MVR-cache improves over vCache: both methods may see similar coarse candidates, but MVR-cache can identify a better Top-1 nearest neighbor.
>
> To validate this, we collect the Top-1 nearest neighbor for each new prompt of the PromptBench dataset using two approaches: one is our two-stage pipeline (single-vector retrieval + MVR reranking), while the other one is to purely use MVR (multi-vector retrieval) to scan the entire cache. The results are very close: Recall@1 = 0.1790 for our pipeline versus 0.1883 for the full MVR scan. This small gap suggests that the single-vector retrieval stage retains most of the real Top-1 nearest neighbors that the full MVR scan can identify, while the MVR reranker can improve the final Top-1 accuracy.
>
> - **Regarding marginal latency reduction**
>
> Thank you for this comment. We agree that the end-to-end latency gains in Table 1 are marginal. Indeed, Figure 4 in Section 6 suggests that these experiments operate in a low-hit-rate regime: in all four datasets, the hit rate remains below 20%, so most prompts still require an LLM call. As pointed out in our response to reviewer dJdE, the LLM call dominates end-to-end latency while MVR-cache only incurs negligible overhead. In addition, as noted in our response to reviewer Tjk6, the vcache’s semantic-caching protocol that we follow is conservative. When a cache hit happens, the new prompt is not inserted into the cache, which can reduce later cache hit opportunities. To mitigate this, we also ran a controlled experiment in which all prompts were inserted regardless of cache hit or miss. Under this setting, the end-to-end latency reduction of MVR-cache becomes more visible:
>
> - SemCacheClassification: MVR-cache 336.95 min vs vCache 378.23 min (10.9% lower)
> - SemCacheSearchQueries: MVR-cache 6062.41 min vs vCache 6168.87 min (1.7% lower)
> - PromptBench: MVR-cache 1229.08 min vs vCache 1258.13 min (2.3% lower)
> - QNLI: MVR-cache 1023.77 min vs vCache 1079.46 min (5.2% lower)
>
> Although the end-to-end latency reduction still appears numerically small, it can still be meaningful in practice. Indeed, by dividing the above cumulative latency by the number of prompts, the average latency reduction per prompt is around 50 ms. As suggested by prior HCI studies [1], users can perceive latency differences well that are below 10 ms. This indicates that the end-to-end latency reduction of MVR-cache is meaningful for improving user experience.
>
> [1] Attig et al. *System Latency Guidelines Then and Now – Is Zero Latency Really Considered Necessary?* EPCE, 2017.
>
> - **Regarding the handling of cases involving extremely long contexts**
>
> Thank you for raising this point. We inspected several long prompts in our datasets, each of which contain around 10k tokens. In these cases, MVR-cache can still produce reasonable segmentations and yield correct cache hits, meaning that each of these prompts and the corresponding cached prompts have identical LLM output. In contrast, vCache either yields cache misses or incorrect cache hits. These examples provide qualitative evidence that MVR-cache can operate effectively on long inputs.
>
> To make this concrete, we include these long prompts and the detailed comparison between MVR-cache and vCache in the following anonymous links:
> + MVR-cache: https://anonymous.4open.science/r/MVR-Cache-E4D2/long_prompts_examples/MVRCache.json
> + vCache: https://anonymous.4open.science/r/MVR-Cache-E4D2/long_prompts_examples/vCache.json
> + Readme: https://anonymous.4open.science/r/MVR-Cache-E4D2/long_prompts_examples/Readme

---

> > ### Author Rebuttal · Reviewer_91RM · 2026-04-04
> >
> > I would like to thank the authors for their detailed rebuttal. The additional data and controlled experiments provide much-needed clarity on the system's actual behavior. I will finalize my score at Weak Accept.

---

### Official Review · Reviewer_Tjk6 · 2026-03-13

**Soundness:** 3
**Presentation:** 3
**Significance:** 3
**Originality:** 3
**Overall Recommendation:** 4
**Confidence:** 3

**Summary:**

MVR-cache improves LLM semantic caching by replacing single-vector cosine similarity with Multi-Vector Retrieval (MVR) and a MaxSim score.

**Compliance With Llm Reviewing Policy:**

Affirmed.

**Final Justification:**

Overall, the rebuttal resolves Q1 and Q2 effectively — particularly Q2, which was my primary empirical concern. Q3 remains partially open. I maintain my current score.

**Key Questions For Authors:**

1. Section 3.4 specifies that the candidate split positions Px consist exclusively of punctuation marks. Semantic boundaries in LLM prompts often fall at other positions (conjunctions, clause boundaries, token-level semantics). No sensitivity analysis is provided for this choice, nor any comparison to word-level or sentence-level candidate sets.

2. The claim "up to 25% higher cache hit rates" refers specifically to SemCacheClassification. On SemCacheSearchQueries, the gain over vCache is marginal.

3. The training pipeline requires ground-truth LLM responses for all training prompts. Semantic caching is most valuable for expensive or rate-limited LLMs (GPT-4, Claude), where labeled responses at training time are costly.

**Limitations:**

See questions.

**Strengths And Weaknesses:**

1. The latency analysis in Table 1 shows that MVR-cache reduces end-to-end inference time compared to vCache despite adding segmentation and multi-vector retrieval overhead.

2. Theorem 3.3 and Lemma 3.4 formally establish that minimizing the class-rebalanced MLE loss increases the separation between the score distributions of matching and non-matching pairs, translating directly to higher cache hit rates under a fixed true-positive guarantee.

---

> ### Author Rebuttal · Authors · 2026-03-31
>
> We thank the reviewer for the comments. Our responses are below.
>
> + **Regarding the sensitivity analysis on the split choice:**
>
> We have now added a sensitivity analysis on the choice of candidate split positions, while keeping the rest of the MVR-cache pipeline fixed. Specifically, we compared four candidate sets on PromptBench:
>
> - keyword-level: punctuation marks and selected keywords (for example, “and”, “or”)
> - token-level: punctuation marks and spaces
> - sentence-level: punctuation marks excluding commas
> - punctuation-level: punctuation marks (default)
>
> The resulting cache hit rates are:
>
> - keyword-level: 3430 / 36369 = 9.43%
> - token-level: 3441 / 36369 = 9.46%
> - sentence-level: 3485 / 36369 = 9.58%
> - punctuation-level: 3555 / 36369 = 9.77%
>
> Notably, even when we gave the RL policy more freedom through larger candidate sets such as keyword-level and token-level boundaries, the final cache hit rates remained very close since punctuation marks are indeed selected most times. This suggests that expanding the candidate set beyond punctuation marks does not lead to a significant performance change in our current setting.
>
> The figures of the cumulative cache hit rate VS increasing number of incoming prompts of these four strategies are available here: https://anonymous.4open.science/r/MVR-Cache-E4D2/Sensitivity_analysis_boundaries/multi_dataset_cache_hit_comparison_01_sensitivity_analysis.png
>
> - **Regarding the marginal improvements:**
>
> Thank you for this observation. We agree the gain over vCache on SemCacheSearchQueries is marginal. However, our further investigation suggests that this is caused by the vcache’s semantic-caching protocol that we follow: once an incoming prompt is successfully hit by some cached prompt, this new prompt is not inserted into cache. As a result, a method yielding higher cache hit rate can actually end up with a smaller set of cached prompts over time, which can in turn reduce later cache hit opportunities. To isolate this effect, we performed an additional controlled experiment in which all prompts are inserted into the cache regardless of cache hits or misses to ensure identical cached prompt sets across different methods. Under this protocol, the improvement of MVR-cache over vCache on all four datasets becomes much clearer:
>
> - SemCacheClassification - MVR-cache (38.82%) VS vCache (29.73%)
> - SemCacheSearchQueries - MVR-cache (8.44%) VS vCache (6.17%)
> - PromptBench - MVR-cache (40.83%) VS vCache (39.01%)
> - QNLI - MVR-cache (46.02%) VS vCache (42.84%)
>
> We provide the corresponding figures of the cumulative cache hit rate VS increasing number of incoming prompts for these controlled experiments at the following link:  https://anonymous.4open.science/r/MVR-Cache-E4D2/Insert_all_samples_results/multi_dataset_cache_hit_comparison_01_insert_all.png
>
> - **Regarding the growth-truth LLM data collection for training**
>
> Thank you for raising this point. We agree that requiring ground-truth LLM responses for training is costly. In light of this, we explicitly keep the training split small for this reason: only 3K training prompts are used as mentioned in Section 4.1.
>
> In addition, our ablation study shows that MVR-cache is data-efficient. As reported in Appendix C (Figure 20), varying the number of training samples leads to almost the same final performance, indicating that increasing the training set provides no significant benefits. Hence, 3K training samples are already sufficient to learn a reasonable segmentation model.

---

> > ### Author Rebuttal · Reviewer_Tjk6 · 2026-04-03
> >
> > Thank you for the detailed rebuttal.
> >
> > On the segmentation search space (Q1): The sensitivity analysis across four candidate sets (keyword/token/sentence/punctuation) is appreciated. The fact that all variants converge to similar hit rates (~9.4–9.8%) reasonably supports the current design choice. This concern is addressed.
> >
> > On the marginal improvements (Q2): The controlled experiment (inserting all prompts regardless of hit/miss) is a compelling addition. The "self-penalization" effect under vCache's protocol is a valid explanation, and the controlled results (e.g., SemCacheSearchQueries: 8.44% vs 6.17%) now show consistent gains across all four datasets. This concern is satisfactorily addressed.
> >
> > On the labeled data cost (Q3): I acknowledge the point that only 3K training samples are needed and that performance is stable across training set sizes. However, the response does not address the cost-benefit tradeoff between labeling effort and caching savings, nor discuss whether semi-supervised or unsupervised alternatives could remove this requirement entirely. This concern is partially addressed.
> >
> > Overall, the rebuttal resolves Q1 and Q2 effectively — particularly Q2, which was my primary empirical concern. Q3 remains partially open. I maintain my current score.

---

> > > ### Author Response · Authors · 2026-04-05
> > >
> > > Thank you for your further clarification on this thoughtful comment. We agree that the cost of obtaining training labels should be weighed against the downstream caching benefit. In our setting, this tradeoff is favorable: the segmentation model is trained with only 3K labeled prompts, while the resulting improvement in cache hit rate yields substantially larger savings at inference time. For example, on our updated SemCacheClassification results, MVR-cache improves the hit rate by about 9% over vCache, which corresponds to about 4.1K fewer LLM calls on the evaluation stream alone—already exceeding the one-time cost of collecting labels for the 3K training samples. Moreover, our cross-dataset generalization results suggest that this labeling cost does not necessarily need to be paid repeatedly for every deployment setting.
> > >
> > > Regarding whether the requirement of labeled training samples can be removed entirely, we agree that this is an important direction. In our framework, supervision is essential to determine whether two prompts lead to equivalent LLM outputs. However, our further investigation suggests that the labels do not have to come from the expensive LLMs but can be produced by smaller alternatives, which is a form of weak supervision. We experimented with using GPT-4o-mini to approximate these labels produced by GPT-4o on the 3K training samples of SemCacheClassification. To accomplish this, we use the generation likelihood of the GPT-4o-mini output as a confidence score, and defer to GPT-4o only when this score falls below a threshold. With the threshold set to -0.002, 80.4% of prompts are labeled by GPT-4o-mini, and the labeling agreement with GPT-4o on these prompts is 97.01%. The remaining 19.6% of prompts can then be labeled directly by GPT-4o. This means that up to 80% of GPT-4o invocations can be avoided, with only very negligible surrogate-label error. These results suggest that the supervision cost can be reduced substantially in practice, even without changing our current training framework. This proxy-oracle strategy has been broadly used in prior work such as LOTUS, which uses cheaper proxy models and defers to a stronger model only when needed [1].
> > >
> > > [1] Patel et al. Semantic Operators: A Declarative Model for Rich, AI-based Data Processing

---

### Decision · Program_Chairs · 2026-04-30

**Decision:**

Accept (regular)

**Comment:**

This paper proposes to optimizing Semantic Caching through two key ways: multi-vector retrieval and learned prompt segmentation. The segmentation model is optimized by reinforcement learning. The proposed approach is helpful for the research in this area. All the reviews were positive. Most review comments were addressed. The AC would like to recommend Acceptance.